# Climate Impacts of Parameterizing Subgrid Variation and Partitioning of Land Surface Heat Fluxes to the Atmosphere with the NCAR CESM1.2

Ming Yin[1], Yilun Han[1], Yong Wang[1], Wenqi Sun[1], Jianbo Deng[1,2], Daoming Wei[1], Ying Kong[3], Bin Wang[1,4,5]

[1]Department of Earth System Science, Ministry of Education Key Laboratory for Earth System Modeling, Institute for Global Change Studies, Tsinghua University, Beijing, 100084 China
[2]Hunan Institute of Meteorological Sciences, Changsha, 410118 China
[3]College of Atmospheric Sciences, Lanzhou University, Lanzhou, 730000 China
[4]State Key Laboratory of Numerical Modeling for Atmospheric Sciences and Geophysical Fluid Dynamics, Institute of Atmospheric Physics, Chinese Academy of Sciences, Beijing, 100029 China.
[5]College of Earth and Planetary Sciences, University of Chinese Academy of Sciences, Beijing, 100029 China.

*Correspondence to*: Yong Wang (yongw@mail.tsinghua.edu.cn)

**Abstract.** All current global climate models (GCMs) only utilize grid-averaged surface heat fluxes to drive the atmosphere, and thus, their subgrid horizontal variations and partitioning are absent. This can result in many simulation biases. To address this shortcoming, a novel parameterization scheme considering the subgrid variations of the sensible and latent heat fluxes to the atmosphere and the associated partitioning is developed and implemented into the National Center for Atmospheric Research (NCAR) Climate Earth System Model 1.2 (CESM1.2). The evaluations show that in addition to the improved boreal summer precipitation simulation over eastern China and the coastal areas of the Bay of Bengal, the longstanding overestimations of precipitation on the southern and eastern margins of the Tibetan Plateau (TP) in most GCMs are alleviated. The improved precipitation simulation on the southern margin of the TP is from suppressed large-scale precipitation, while that on the eastern edge of the TP is due to decreased convective precipitation. Moisture advection for precipitation production is blocked toward the southern edge of the TP, and the anticyclonic moisture transport anomaly over northern China extends westward, suppressing the development of local convection on the eastern edge of the TP. The altered large-scale circulation in the lower atmosphere due to anomalous heating/cooling in the planetary boundary layer is responsible for the change in moisture transport. The performance of other key variables (e.g., surface energy fluxes, clouds and 2 m temperature) is also evaluated among the default CESM1.2, the new scheme, and the scheme stochastically allocating the subgrid surface heat fluxes to the atmosphere (i.e., without subgrid partitioning included). This study highlights the importance of subgrid surface energy variations and partitioning to the atmosphere in the simulation of the hydrological and energy cycles in GCMs.

## 1 Introduction

The importance of land surface heterogeneity has been proven through many observational and modeling studies (e.g., Taylor et al., 2007; Lothon et al., 2011; Rochetin et al., 2017; Wang et al., 2017). The variability of the surface heat fluxes caused by the heterogeneity of the surface properties is crucial to the turbulence in the planetary boundary layer (PBL), as well as the evolution of large-scale atmospheric circulation and clouds (Rieck et al., 2014; Lee et al., 2019). In most global climate models (GCMs), confined by the horizontal resolution (~100–200 km), the subgrid surface heat fluxes to the atmosphere are averaged out, thus degrading the simulation of convection and PBL processes. This is one of the causes of many precipitation simulation errors in GCMs, such as the bias of the rainfall intensity spectrum (e.g., Dai, 2006; O'Brien et al., 2016; Na et al., 2020; Wang et al., 2021a) and the unrealistic precipitation over the Indian summer monsoon region (e.g., Waliser et al., 2012; Wang et al., 2018) and the eastern and southern parts of the steep Tibetan Plateau (TP) (e.g., Zhou et al., 2021).

The land surface energy balance involves many biophysical and biogeochemical processes (Lee et al., 2011; Liu et al., 2014; Duveiller et al., 2018; Chakraborty and Lee, 2019; Liu et al., 2022), which are closely related to surface properties. For instance, forests dissipate sensible heat to the PBL more efficiently than open landscapes (Rotenberg and Yakir, 2010; Wei et al., 2021), and the increase in vegetation density has been found to favor the release of latent heat rather than sensible heat during the past three and a half decades (Forzieri et al., 2020). The different performance of the energy terms also suggests the potential importance of surface energy partitioning. However, the grid-scale surface heat fluxes to the atmosphere are rudimentarily treated by calculating the weighted averages within each grid cell in all GCMs. This simplified approach inevitably hampers our understanding of small-scale land-atmosphere feedback, which is among the critical processes in efforts to predict future climate change through GCMs (Miralles et al., 2019; Forzieri et al., 2020).

To incorporate the subgrid horizontal variations in the surface heat fluxes to the atmosphere resulting from land cover heterogeneity, a recent study (Sun et al., 2021) proposed a parameterization using stochastic sampling and tested it in the National Center for Atmospheric Research (NCAR) Climate Earth System Model 1.2 (CESM1.2). It was found that this scheme improved the boreal summer precipitation simulation over eastern China. However, Sun et al. (2021) did not comprehensively assess the performance of other variables, and another important limitation is that the simulated summer precipitation on the southern and eastern margins of the TP, similar to most GCMs, is still overestimated compared to observations. The simulation of the TP, which plays an important role in controlling the Asian and global climate, is a longstanding challenge for all of the current GCMs (Mueller and Seneviratne, 2014; Ma et al., 2015). These difficulties arise from the heterogeneity of the underlying surface, the complex terrain, and the sparse observation data used for constraints (Zhou et al., 2019; Liu et al., 2021). All of these factors make it difficult for the existing parameterization schemes to accurately reproduce complex subgrid-scale processes, resulting in degradation of the simulation in the TP region.

In the Sun et al. (2021) scheme, although the subgrid surface heat fluxes to the atmosphere are parameterized via stochastic sampling and internally multiple calls of the PBL and convection schemes, the underlying relationship between the subgrid heat fluxes is neglected. The conversion of the surface available energy into latent and sensible heat fluxes on a subgrid scale

exerts a strong control on global water and energy cycles (Pitman, 2003; Tang et al., 2014; Wang et al., 2021b) by regulating land-atmosphere feedback, especially in regions with complicated land surface features, such as the TP and its surrounding areas (Pielke et al., 2001; Findell et al., 2011; Forzieri et al., 2018, 2020). As the next logical step, in this study, the Sun et al. (2021) parameterization is updated by taking the partitioning between the subgrid sensible and latent heat fluxes into account. It is highly desirable to alleviate the precipitation simulation biases in the TP region through this modification. Given that only the simulated precipitation by the Sun et al. (2021) scheme was investigated, its performance on the simulations of other variables such as grid scale surface energy fluxes, clouds and 2 m temperature is evaluated thoroughly in this study along with that in the modified parameterization.

The manuscript is organized as follows. Section 2 briefly describes the Sun et al. (2021) parameterization scheme and the modifications, CESM and experiments, and the observation and reanalysis datasets. The evaluations of the two schemes based on observations and reanalyses are presented in Sect. 3. The uncertainties are discussed in Sect. 4, while the conclusions are given in Sect. 5.

## 2 Methodology

### 2.1 CESM and Subgrid Heat Flux Scheme

To compare with Sun et al. (2021), the GCM used in this study is the NCAR CESM1.2. The atmospheric component is the Community Atmosphere Model, version 5 (CAM5). The land model is the Community Land Model, version 4 (CLM4). The spatial land surface heterogeneity in the default CLM4 is represented as a nested subgrid hierarchy in which the grid cells are composed of multiple land units, snow/soil columns, and plant functional types (PFTs) (Oleson et al., 2010). All of the fluxes to and from the surface, including the heat fluxes, are defined at the PFT level. Since the subgrid heat fluxes exported to the CAM5 are weighted averages and their weights depend on the fractional coverage of each PFT within the grid cell, the subgrid variations in the land surface fluxes are missing during the land-atmosphere coupling process (Sun et al., 2021).

To consider the influences of the heterogeneity of the subgrid heat fluxes to the atmosphere in CESM1.2, a parameterization scheme was developed and implemented in CLM4. This scheme established the truncated normal distributions of the subgrid sensible and latent heat fluxes independently within the grid cell at each time step. The probability density function (PDF) of subgrid sensible and latent heat flux in a given grid cell was calculated by

$$f(x|\bar{F}, \sigma, F_{min}, F_{max}) = \frac{\frac{1}{\sigma}\phi(\frac{x-\bar{F}}{\sigma})}{\Psi\left(\frac{F_{max}-\bar{F}}{\sigma}\right) - \Psi\left(\frac{F_{min}-\bar{F}}{\sigma}\right)}, x \in [F_{min}, F_{max}] \tag{1}$$

where $\bar{F}$ is the weighted average value of all subgrid heat fluxes, $\sigma$ is the standard deviation, $F_{min}$ and $F_{max}$ are the minima and maxima of the subgrid heat fluxes respectively, and $\phi$ and $\Psi$ are the PDF and the cumulative distribution function (CDF) of the standard normal distribution, respectively. $N$ (i.e., the maximum number of PFTs coexisting in the grid cell) samples of sensible heat fluxes and $N$ samples of latent heat fluxes were independently and randomly paired with each other to drive $N$

independent groups of the PBL and the deep convection parameterization schemes in CAM5. The outputs from these $N$ calls

of the schemes were then averaged with equal weights as the inputs of the other schemes.

The stochastic sampling implicitly parameterized the uncertainties of the PBL and convection processes to a certain degree. As stated in Sun et al. (2021), using the sampled fluxes from a statistical distribution rather than the fluxes directly from individual PFTs can represent the mix of subgrid fluxes from horizontally mixed land cover types in reality. Moreover, the distribution of the sampled subgrid surface heat fluxes based on the assumed normal distribution resembles the distribution of

realistic subgrid PFT heat fluxes within the grid cell in long-term statistics. As shown in Fig. 1 for the sensible heat flux, over the grid cells with 16 and 8 PFTs, the two distributions are highly consistent, in terms of mean value, variance and skewness. The latent heat flux has similar results (figure not shown). Given that those grid cells are stochastically selected and cover different climatic regimes (Fig. S1), the assumed normal distribution works well and thus the sampled samples can represent the realistic features for climate simulation.

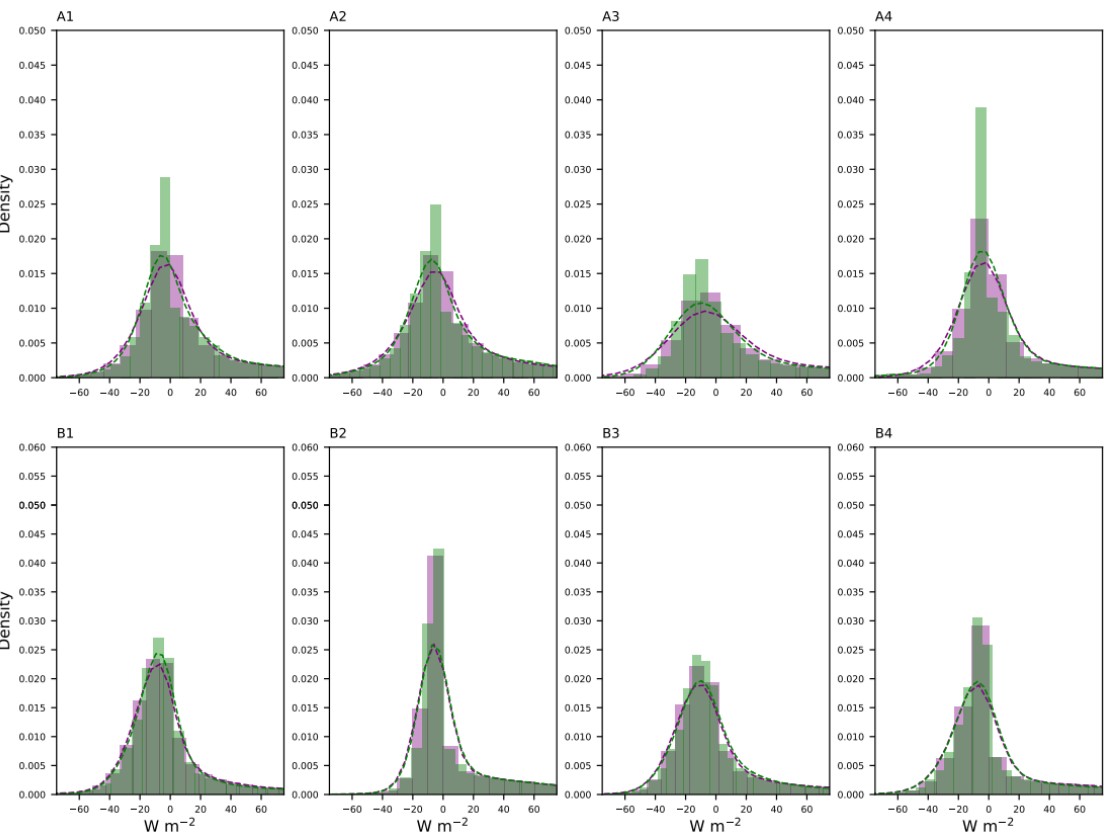

**Figure 1. The histogram and Gaussian kernel density estimate (KDE) (dashed line) of the sensible heat fluxes at the PFT in the eight grid cells with 16 (top row) and 8 (bottom row) PFTs, respectively. Green histograms and KDE estimates show the distribution of realistic sensible heat fluxes at the PFT within each grid cell, while purple histograms and KDE estimates show the distribution of sampled sensible heat fluxes based on the assumed normal distribution.**

Note that the surface energy balance closure at the grid scale is not affected by the stochastic sampling method. The surface energy balance has been closed at the grid scale in the default land-atmosphere coupling way. Therefore, the stochastic sampling at the subgrid scale based on the truncated normal distributions with the mean values equal to the default grid averages calculated by the weighted fluxes on each PFT within the grid cell (Fig. 1) can assure that the grid-scale surface energy balance is closed as well in the long-term statistics, although at a given time step this might be broken up.

## 2.2 Modified Subgrid Heat Flux Scheme

In the stochastic scheme proposed by Sun et al. (2021), the subgrid sensible heat and latent heat fluxes were individually and randomly selected from their truncated normal distributions, without considering the underlying relationship between these two energy terms. However, we can compute the correlation coefficients between the subgrid sensible and latent heat fluxes within each grid cell at every time step (i.e., 30 mins) using the following equation:

$$r = \frac{\sum_{i=1}^{n} w_i (F_{SH_i} - \bar{F}_{SH})(F_{LH_i} - \bar{F}_{LH})}{\sigma_{SH} \sigma_{LH}} \qquad (2)$$

where $n$ is the number of PFTs within a grid cell in the land model; $w_i$ is the area percentage of each PFT within the grid cell; $F_{SH_i}$ and $F_{LH_i}$ are the subgrid surface sensible and latent heat fluxes of each PFT, respectively; $\bar{F}_{SH}$ and $\bar{F}_{LH}$ are the weighted averages of the subgrid sensible and latent heat fluxes in one grid cell, respectively; and $\sigma_{SH}$ and $\sigma_{LH}$ are the standard deviations of the subgrid sensible and latent heat fluxes in one grid cell, respectively. The correlation coefficients vary with time. Figure 2a shows the annual mean distribution of the energy partitioning between the sensible heat and latent heat fluxes at the subgrid scale. There are negative correlations at low latitudes in the Northern Hemisphere (NH) and most of the Southern Hemisphere (SH), whereas in the middle and high latitude regions in the NH and on the TP, most of the regions have positive correlations. The spatial patterns of the June-July-August (JJA) mean and December-January-February (DJF) mean correlation coefficients are given in Fig. 2b&c. In boreal summer, the sensible and latent heat fluxes in most regions of the world are negatively correlated, except for the TP, Greenland, the central US, and southern Australia (Fig. 2b). In boreal winter, the global distribution is similar to that of the annual mean, showing larger magnitudes of the positive correlation coefficients and smaller magnitudes of the negative correlation coefficients (Fig. 2c). The regions where the correlation coefficients are positive in both summer and winter are mainly located in high latitudes and altitudes. In summer, owing to the melting of snow, latent heat flux increases accordingly as the land surface gains more water for evaporation, and sensible heat flux increases synchronously from enhanced surface net radiation due to increased incoming solar radiation and reduced snow albedo. For winter, decreased solar radiation and increased snow cover reduce both sensible and latent heat fluxes.

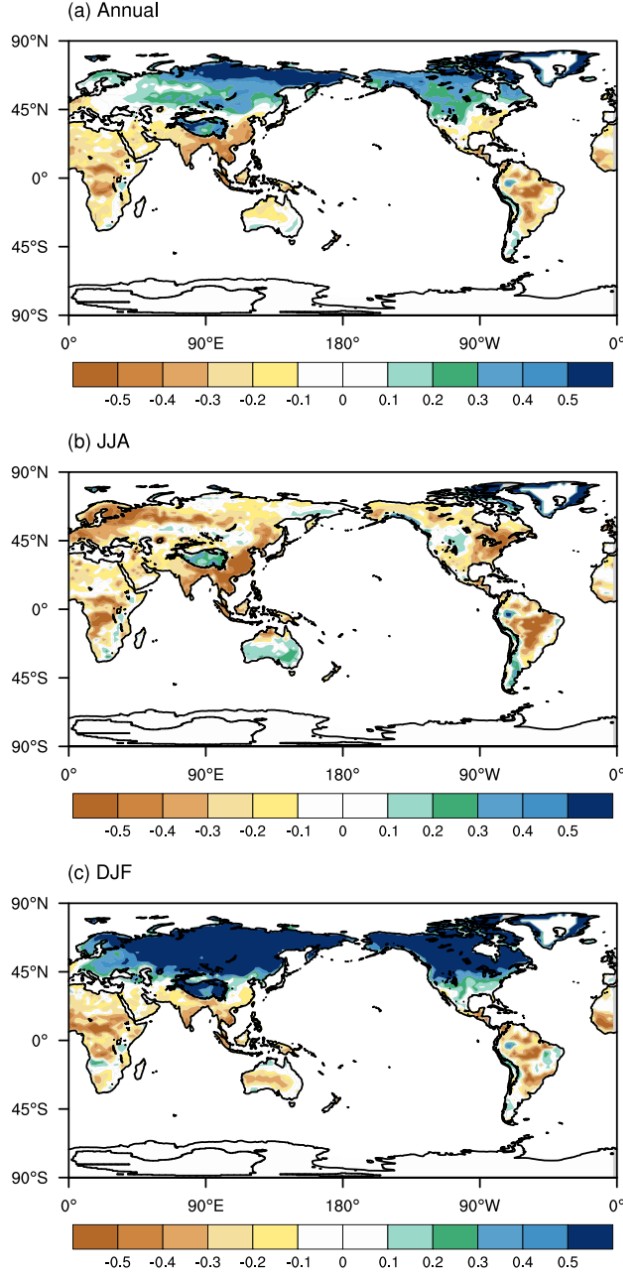

**Figure 2. Spatial distribution of (a) annual, (b) JJA (June-JulyAugust) and (c) DJF (December-January-February) mean correlation coefficients *r* between the subgrid surface sensible heat and latent heat fluxes in the EXP_COR simulation.**

Based on the notion that subgrid energy partitioning between the surface and the PBL is important in the land-atmosphere coupling process, several improvements are made to the Sun et al. (2021) scheme in this study. Two simplified methods are developed.

(1) Arrange the randomly selected $N$ subgrid sensible heat (SH) fluxes and $N$ subgrid latent heat (LH) fluxes in each grid cell from largest to smallest and use the $N$ pairs of matching sensible and latent heat fluxes to drive the atmosphere independently. That is, a large (small) SH flux corresponds to a large (small) SH flux.

(2) Arrange the randomly selected $N$ subgrid sensible heat fluxes from largest to smallest and arrange the $N$ latent heat fluxes from smallest to largest in each grid cell. Then, the $N$ pairs of matching sensible and latent heat fluxes are used to drive the atmosphere independently. That is, a large (small) subgrid SH flux corresponds to a small (large) subgrid LH flux.

Which one is used for a given grid cell depends on the time-varying correlation coefficient $r$. If the correlation coefficient $r$ in the grid cell is positive, the PBL and convection parameterizations are driven using the heat fluxes derived in method one. Otherwise, the heat fluxes selected using method two will be passed to the atmosphere. The arithmetic mean of the outputs from $N$ calls of the PBL and the convection parameterizations is input into the other following schemes. Given that the surface energy balance closure at the grid scale is not affected by the stochastic sampling method, the follow-up collocation of the sampled sensible and latent heat fluxes according to their correlation coefficient does not break up this rule. This is because this process does not alter the sampled subgrid values just arranging them in a given sequence.

## 2.3 Experiments

Three Atmospheric Model Intercomparison Project (AMIP)-type experiments with a finite volume dynamical core at a horizontal resolution of 1.9°×2.5° (~2°) and 30 vertical levels from the surface to 3.6 hPa were conducted using observed climatological (1982–2001 mean) monthly sea surface temperature and sea ice extent data (Stone et al., 2018). One control simulation (CTL) uses the standard CESM1.2, another experimental simulation (EXP) uses the Sun et al. (2021) parameterization in CESM1.2 (also the same as the EXP run in their study), and the third improves the EXP run using the modifications described in Sect. 2.2 (EXP_COR). All of the simulations were run for six years, with the first year discarded as the spin-up stage. The value of $N$ in each grid cell was fixed to 16, which equals the maximum number of PFTs ever coexisting on a single column in the land model, although different grid cells have different numbers of PFTs (Sun et al., 2021). As noted by Sun et al. (2021), further increasing $N$ has negligible impacts on the model performance compared with setting $N$ to 16 and enhances computational loading instead.

## 2.4 Observations and Reanalyses

To evaluate the model performance, the simulation results are compared with the available observation and reanalysis datasets. The Tropical Rainfall Measuring Mission (TRMM; Huffman et al., 2014) observations (0.25°×0.25°) and the Modern-Era Retrospective Analysis for Research and Applications, Version 2 (MERRA-2; Gelaro et al., 2017) reanalysis (0.5°×0.625°) are used for precipitation. The other datasets include surface radiative fluxes from the Clouds and the Earth's Radiation Energy Systems (CERES) Energy Balanced and Filled (1.0°×1.0°) (EBAF; Loeb et al., 2012), sensible heat and latent heat fluxes from the Global Land Data Assimilation System Version 2.1 (GLDAS-2.1) Noah monthly data (1.0°×1.0°) (Rodell et al., 2004) and

2 m air temperature from the Climatic Research Unit with a 0.5° resolution (CRU; Harris et al., 2020). For consistency, all of the observation/reanalysis datasets are regridded to the same grid size as CAM5.

## 3 Results

Sun et al. (2021) found that the improved precipitation simulation with the parameterization of subgrid surface heat fluxes to the atmosphere is most prominent for boreal summer and in Asia. In this study, the following analyses are still mainly focused on boreal summer because one improvement we expected in the new scheme is the alleviation of the overestimated summer precipitation on the southern and eastern margins of the Tibetan Plateau (TP) as shown in most GCMs. Moreover, a thorough evaluation of the two parameterizations on simulated climate variables at the global scale and for four seasons is performed. Their global annual and seasonal statistics are given in Sect. 3.3.

### 3.1 Precipitation

Sun et al. (2021) (i.e., the EXP run) improved the simulation of the summer precipitation over eastern China and the coastal areas of the Bay of Bengal (Fig. 3b-d), which was attributed to altered vertical diffusion and convection. The simulated precipitation over Arabia and Indonesia is improved as well while that over the southeastern US is degraded. In particular, it still produces excessive precipitation on the eastern and southern margins of the TP. After taking the subgrid energy partitioning into account in the EXP_COR run, the overall performance in terms of the globally averaged root mean square error (RMSE) and the spatial correlation coefficient (COR) is comparable to that of the EXP run (Fig. 3d&f). In addition to the improved boreal summer precipitation simulation over eastern China and the coastal areas of the Bay of Bengal, the longstanding overestimations of precipitation on the southern and eastern margins of the TP are alleviated by up to -2.5 mm d$^{-1}$ (Fig. 3b-f), although the simulated precipitation is still excessive. Over other regions such as southern China, the Middle East and Indonesia, there are some slight degradations in the EXP_COR run compared to the EXP run.

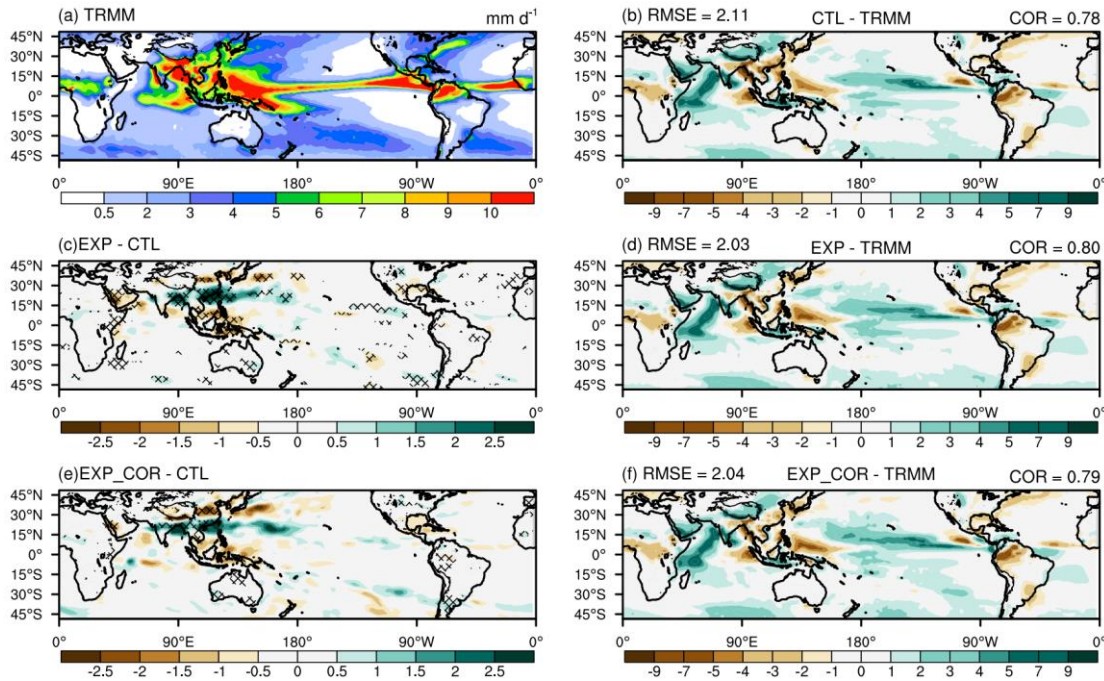

**Figure 3. Spatial distributions of the JJA (June-July August) mean precipitation for (a) TRMM, the biases of (b) CTL, (d) EXP, and (f) EXP_COR with respect to TRMM, and the differences between EXP and CTL (c) and between EXP_COR and CTL (e). The crossed areas are significant at the 95% level. The regionally averaged spatial correlation coefficient (COR) and root mean square error (RMSE) are calculated at the top of (b), (d) and (f).**

Figure 4 zooms in on the region (20-50°N, 75-125°E) where the simulated precipitation exhibits obvious improvements in the EXP_COR run. In the CTL run, the wet bias over the southern margin of the TP can exceed 11 mm d$^{-1}$ while that over the eastern margin of the TP is approximately 7 mm d$^{-1}$. Additionally, seen in other CMIP5&6 models, the biases there are much larger than those in the rest of the world (Fig. 3) (Su et al., 2013; Yu et al., 2015; Zhu and Yang, 2020; Lun et al., 2021). In contrast, in the EXP_COR run, the reduced biases over these two regions can be as much as 2.5 mm d$^{-1}$ accounting for a reduction of approximately 25%, especially over the southern margin of the TP. Given that there are many causes (e.g., unrealistic water vapor advection and the absence of subgrid topographic effects) resulting in the severe overestimation of precipitation along the TP, the improvement in this study, to some extent, is impressive. The regionally averaged RMSE decreases from 4.51 in the CTL run and 4.07 in the EXP run to 3.71 in the EXP_COR run, and the COR increases from 0.48 in the CTL run to 0.60 in both the EXP and EXP_COR runs.

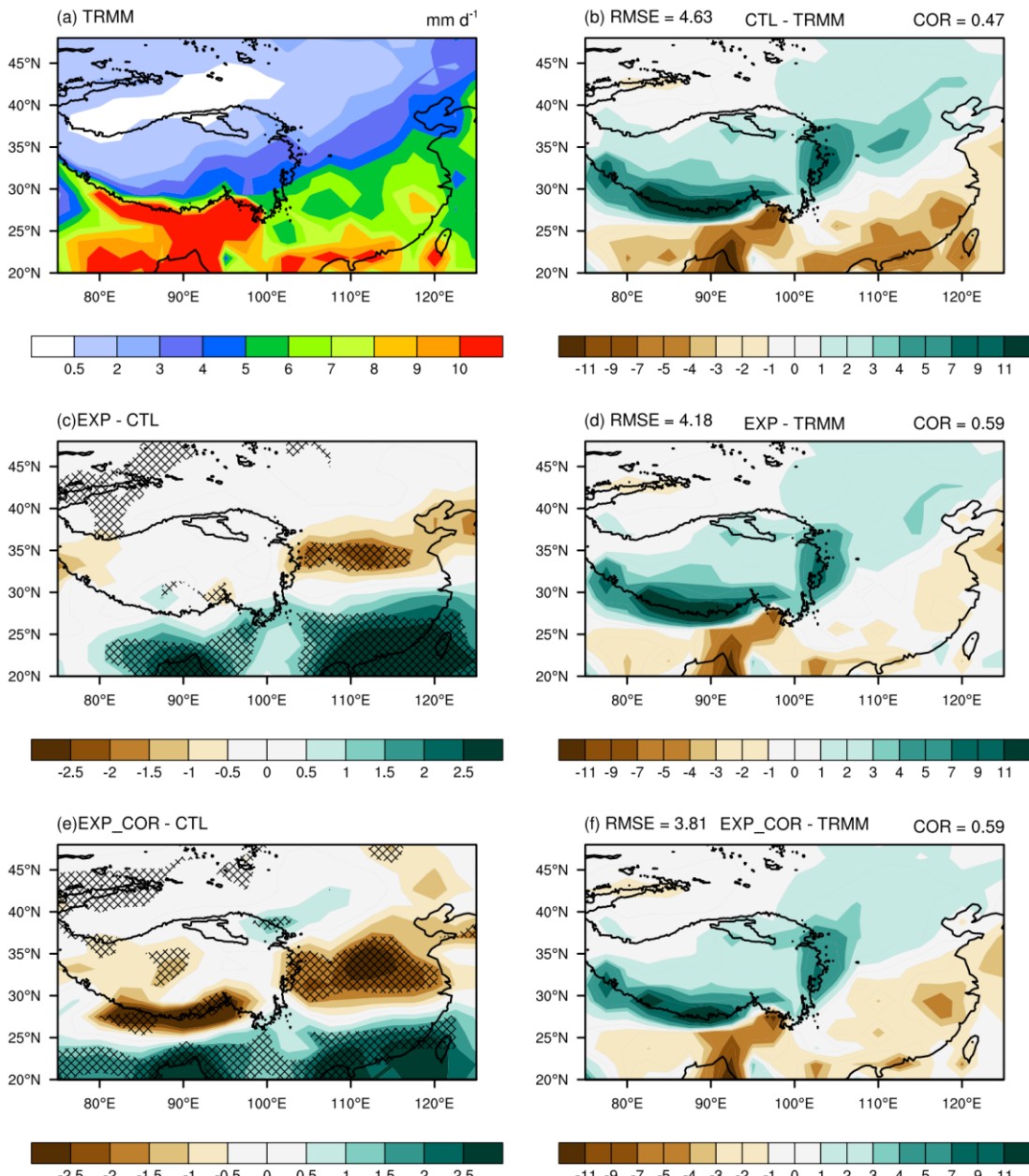

**Figure 4. Same as Fig. 3 but focusing on the study area (20-50°N, 75-125°E). The regionally averaged spatial COR and RMSE are calculated at the top of (b), (d) and (f).**

The total precipitation in the model consists of convective and large-scale precipitation. Their contributions are analyzed accordingly. Figures 5a&c show that in the EXP run, compared with the CTL run, both large-scale precipitation and convective precipitation slightly increase on the southern border of the TP. On the eastern border of the TP, large-scale precipitation increases, and convective precipitation decreases. In contrast, in the EXP_COR run, large-scale precipitation is significantly

suppressed on the southern fringe, and both large-scale precipitation and convective precipitation are reduced on the eastern margin.

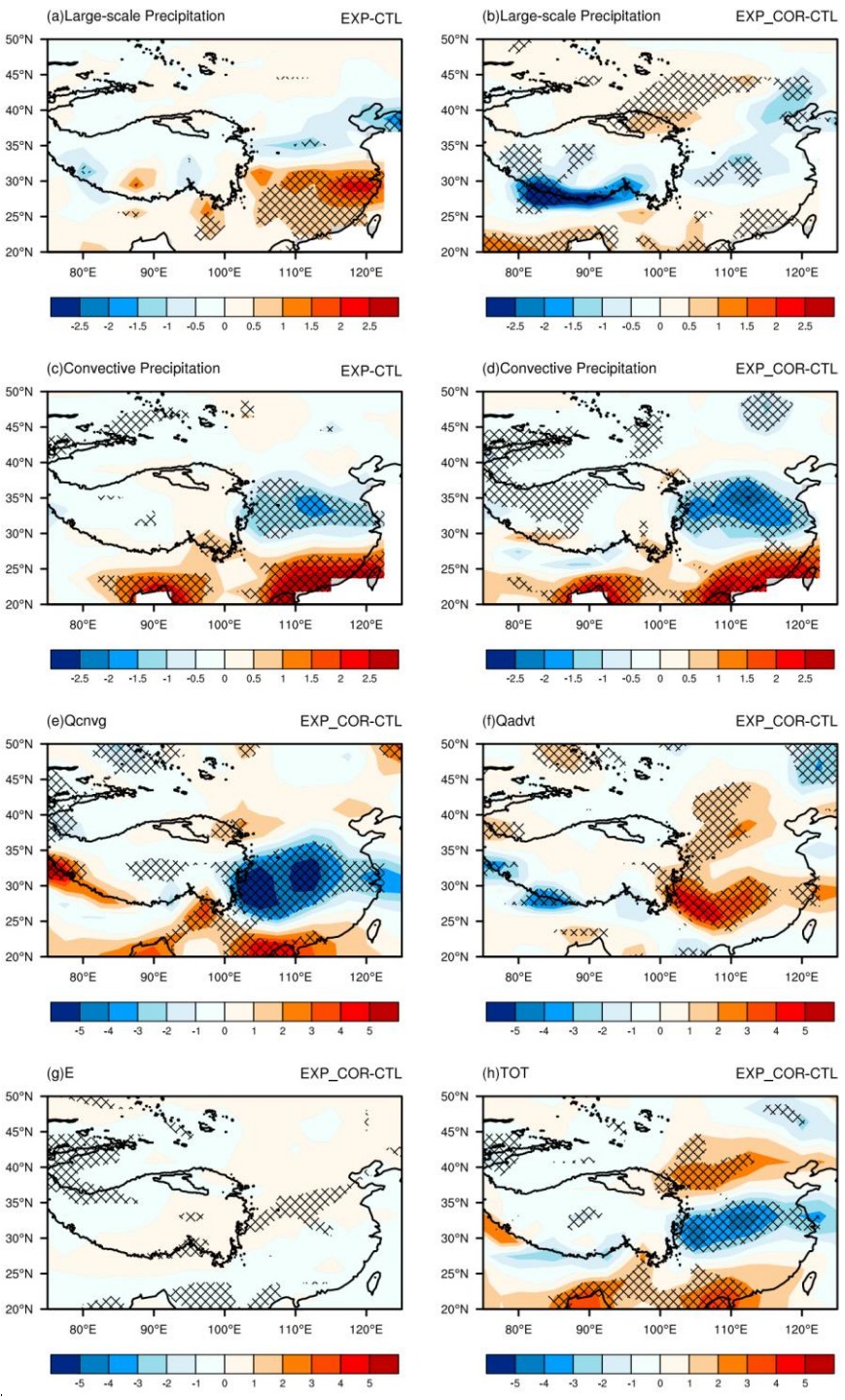

A moisture budget analysis widely used in previous studies (Gao et al., 2017; Wang et al., 2016) is conducted to examine the causes of precipitation changes. Following Sun et al. (2021), the atmospheric water vapor budget equation is given below:

$$\frac{\partial W}{\partial t} = \left(-W\nabla \cdot \vec{V}\right) + \left(-\vec{V} \cdot \nabla W\right) - P + E \tag{3}$$

where $P$ is precipitation and $E$ is evaporation. $W$ is the column-integrated moisture given by $\int_{P_{top}}^{P_{bot}} q\,dp/g$, in which $q$ is the specific humidity, $P_{top}$ and $P_{bot}$ are the top and surface pressures, respectively, and $g$ is the acceleration due to gravity. The vector $\vec{V}$ with units of m s$^{-1}$, given by $W^{-1}\int_{P_{top}}^{P_{bot}}(q\vec{u})dp/g$, represents the total horizontal moisture transport normalized to the column-integrated moisture, where $\vec{u}$ is the horizontal wind vector. The first term on the right-hand side of Eq. (3) is the moisture convergence $Q_{cnvg}$, and the second term is the moisture advection $Q_{advt}$. The tendency of the term $\frac{\partial W}{\partial t}$ on the left-hand side of Eq. (3) is negligible for seasonal averages of multiple years.

Compared with the CTL run, moisture convergence weakens on the eastern edge of the TP, while moisture advection increases in the EXP_COR run (Figs. 5e&f). On the southern edge of the TP, moisture advection decreases, and moisture convergence slightly increases. Overall, consistent with the change of total precipitation, the total water vapor contributions decrease on the eastern and southern edges of the TP (Fig. 5h). We note that the spatial pattern of $Q_{cnvg}$, changes in the EXP_COR run relative to the CTL run, resembles that in the EXP run (Figs 4d&f in Sun et al., 2021), which is linked with the changes in the heating rate due to vertical diffusion in the PBL caused by the subgrid variations in land surface heat fluxes. In comparison with the EXP run, the negative moisture convergence anomaly is further aggravated, and the positive bias of moisture advection on the eastern margin of the TP is smaller (Sun et al. 2021). The negative maximum of the total contribution thus shifts westward to the eastern margin of the TP. Overall, moisture convergence dominates the change of precipitation on the eastern border of the TP (Fig. 5e&h). On the southern edge of the TP, the main term contributing to precipitation changes is due to the reduced moisture advection (Fig. 5f&h).

The causes of the altered moisture convergence and advection are illustrated in Figs. 6 and S2 where the MERRA-2 reanalysis is included for reference. In the EXP run, the subgrid variations of the land surface heat fluxes increase (decrease) PBL heating over southern (northern) China (Fig. 6a). After further taking the partitioning of subgrid surface heat fluxes into account, the increase (decrease) in the heating rate over southern (northern) China is strengthened (Fig. 6b). Therefore, destabilization (stabilization) in the lower atmosphere is further enhanced, promoting (suppressing) the development of local convection. Lower (higher) sea level pressure (SLP) anomalies over southern (northern) China are generated in the EXP_COR run than in the EXP run. In particular, compared with the EXP run, the anomalous high SLP over northern China slightly extends to the south and the anomalous low SLP over southern China retreats (Fig. 6d-h). The anomalous anticyclone over northern China expands accordingly, which engenders decreased precipitation on the eastern border of the TP and a slight dry bias over

southern China. Similar to the EXP run, convective precipitation dominates the changes of total precipitation over eastern China and the eastern margin of the TP in the EXP_COR run. In the EXP run, negative SLP anomalies appear along the Bay of Bengal leading to cyclonic moisture transport from the ocean in the south (Fig. 6e). As a result, excessive moisture is transported to the southern edge of the TP producing excessive rainfall there. In contrast, in the EXP_COR run (Fig. 6g), the easterly anomaly along 25° N-30° N partly blocks moisture transport from the ocean in the south to the southern margin of the TP, and therefore, the decrease of large-scale precipitation dominates the change of precipitation simulation on the southern margin of the TP.

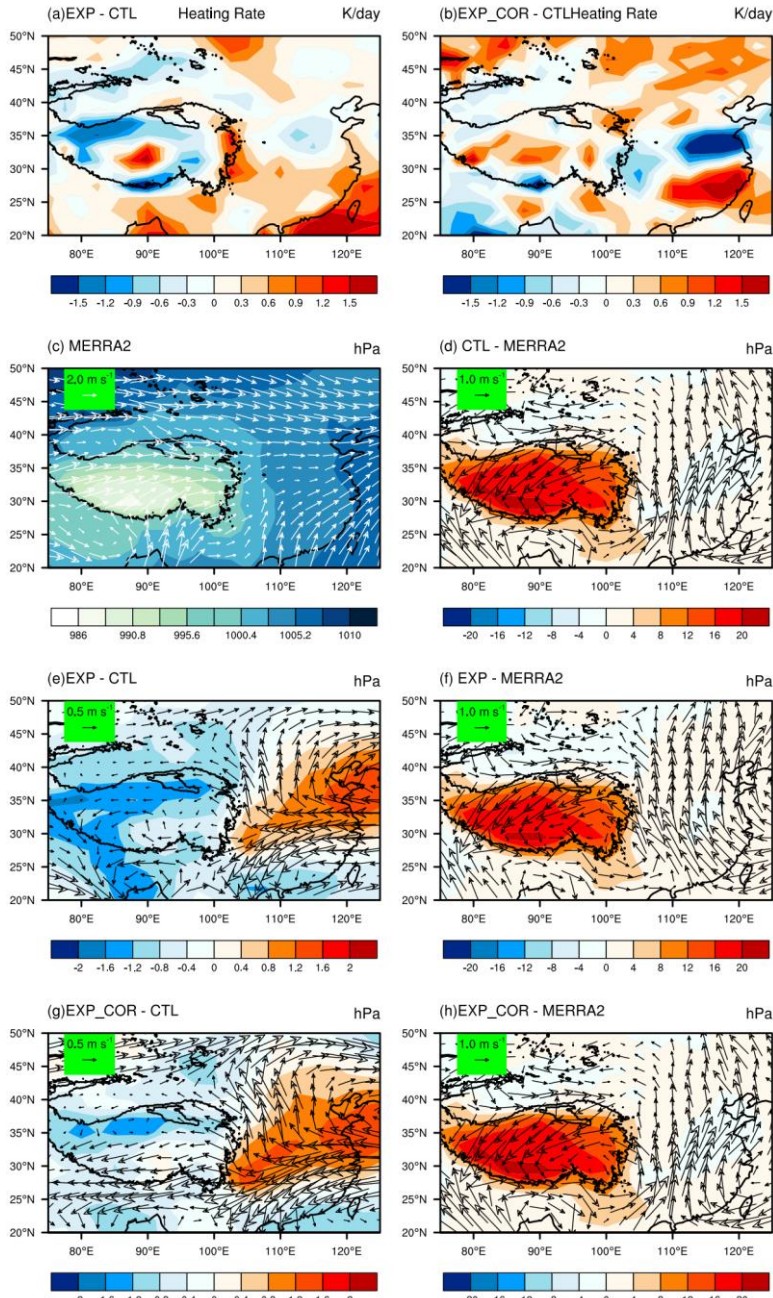

**Figure 6. Spatial distributions of the differences of JJA-mean PBL heating (a) between EXP and CTL, and (b) between EXP_COR and CTL, JJA-mean SLP superposed by the vector $\vec{V}$ from (c) MERRA-2, and the differences (d) between CTL and MERRA-2, (e) between EXP and CTL, (f) between EXP and MERRA2, (g) between EXP_COR and CTL, and (h) between EXP_COR and MERRA2. The vector $\vec{V}$ is defined in Eq. (3).**

**3.2 Surface Heat Fluxes, Clouds and 2 m Air Temperature**

The above analysis indicates that the precipitation simulation is improved through the adjustment of large-scale atmospheric circulation in the lower atmosphere, which is highly linked with grid-scale surface heating/cooling (Sun et al., 2021). The following analyses will evaluate the performance of other variables such as surface energy budgets, clouds, and 2 m air temperature in JJA globally.

     The evaluations of the latent heat flux simulation are shown in Fig. 7. In those regions with large latent heat fluxes in GLDAS

(e.g., the eastern US, northern South America, eastern China, etc.), the simulated values are generally underestimated in the CTL run, while in the regions with relatively small latent fluxes (e.g., the Arabian Peninsula, the Sahara Desert, and the northwestern TP, etc.), CTL tends to overestimate their values. Overall, the three simulations have similar distributions and comparable CORs. The EXP run has the smallest RMSE, followed by the EXP_COR run. Specifically, with subgrid variations of surface heat fluxes incorporated, compared with the CTL run, the EXP run reduces the biases over the central US,

northeastern China and southeastern Russia (Fig. 7b, c&e). Also, the positive biases over southern China, the northwestern TP, the Arabian Peninsula, the Sahara Desert, northwest India and along the Bay of Bengal are reduced, although some degradations along 60°N over the Eurasian continent and southeastern Australia are found. In the EXP_COR run, it inherits most improvements in the EXP run (Fig. 7b, d&f). Furthermore, the biases on the southern and eastern margins of the TP and along 60°N in both the CTL and EXP runs are reduced (Fig. 7b-f). However, in the regions where the correlation coefficients

$r$ are small (Australia, the Arabian Peninsula, the Sahara Desert, etc.) (Fig. 2a), there are no improvements noticed in the EXP_COR run, or even the simulation is degraded.

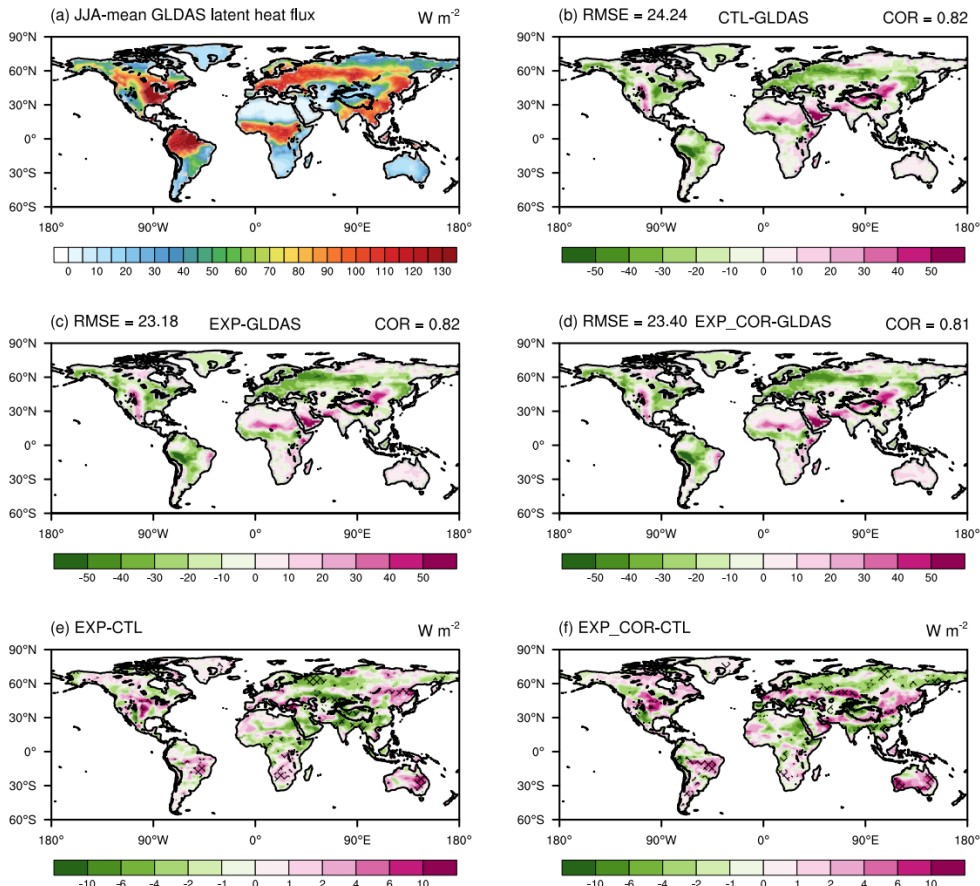

**Figure 7. Spatial distributions of the JJA-mean latent heat flux in (a) GLDAS (upward positive), the biases of (b) CTL, (c) EXP, and (d) EXP_COR with respect to GLDAS, and the differences between (e) EXP and CTL, and between (f) EXP_COR and CTL. The crossed areas are significant at the 95% level. The averaged spatial COR and RMSE for the three simulations are given in (b - d).**

For the sensible heat flux simulation (Fig. 8), in general, the simulated sensible heat fluxes in CTL are underestimated and overestimated over those regions with large and small values in GLDAS, respectively. The RMSE in the EXP run is the smallest among the three experiments, which have comparable correlations. In comparison with the CTL run, the EXP run slightly reduces the positive biases in Europe but degrades the underestimation in Australia (Fig. 8b, c&e). Other improvements can be found in the central US, the Sahara Desert, the Arabian Peninsula, northwestern India, eastern China, the TP, and southeastern Russia. On top of the EXP run, EXP_COR further alleviates the overestimation along 45°N-60°N over the Eurasian continent where sensible heat fluxes and latent heat fluxes are highly correlated in this region (Fig. 2b). The positive changes over the southern and eastern margins of the TP in the EXP_COR run are more significant than those in the EXP run (Fig. 8e&f). Nonetheless, we note some degradations from EXP to EXP_COR (e.g., over southern China).


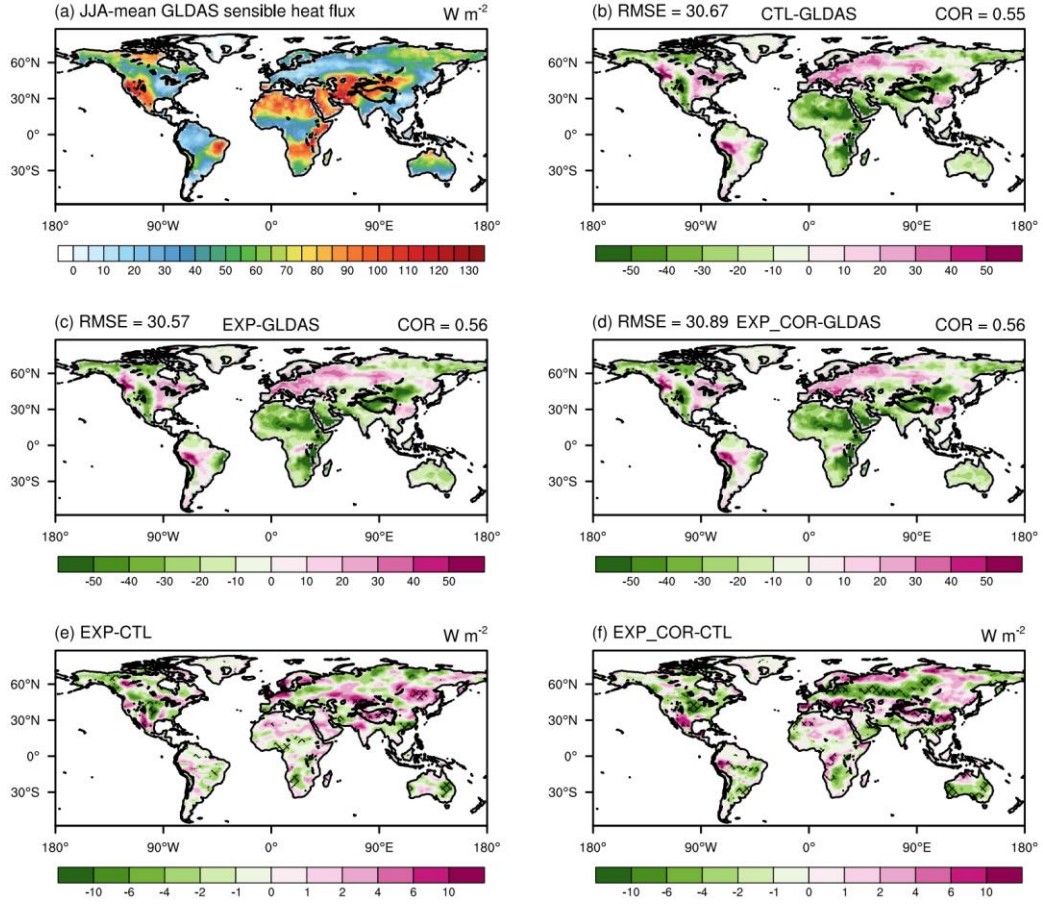

**Figure 8. Spatial distributions of the JJA-mean sensible heat flux in (a) GLDAS (upward positive), the biases of (b) CTL, (c) EXP, and (d) EXP_COR with respect to GLDAS, and the differences between (e) EXP and CTL, and between (f) EXP_COR and CTL. The crossed areas are significant at the 95% level. The averaged spatial COR and RMSE for the three simulations are given in (b - d).**

As indicated in Sect. 3.1, large-scale atmospheric circulation in the lower atmosphere and local convection are altered as PBL

heating changes affect clouds as well. The changes in clouds in turn influence surface radiation and thus surface heat fluxes.

The cloud properties affecting the cloud radiative effects include their macrostructures (e.g., fraction, top and base heights,

and vertical overlap) and microphysical properties (e.g., particle size distribution and geometric configuration, cloud phase

and water condensation). As shown in Fig. 9c, the EXP_COR run reduces low clouds over northern China and southeastern

Russia, and increases them over southern China, the central US and along 60°N in comparison with the CTL run. The EXP

run has a similar pattern of changes but with smaller magnitudes compared with the EXP_COR run (Fig. 9a-c). Low clouds

reflect a larger amount of incoming solar radiation and emit longwave radiation at relatively high temperatures, so they exert

an overall net cooling effect on the Earth (Klein and Hartmann, 1993; Hartmann, 1994). Compared with the CTL run, the

middle and high clouds on the TP are dramatically decreased in the EXP_COR run, and the land surface is warmed because it

gains more net solar radiation. Especially for high clouds, the decrease in the EXP_COR run is much larger than that in the EXP run.

The simulations of the total cloud water path (vertically integrated cloud liquid and ice water content, CWP) are shown in Fig. 10d-f. A higher cloud water content reflects more solar radiation. The EXP run increases the total CWP over southern China, central Africa, the central US, southeastern Australia and along the Bay of Bengal. The CWP is decreased over northern China,

the TP, and southeastern Russia. In the EXP_COR run, the simulated CWP is further decreased on the TP and over northern China, while it is increased in southern China and along 45°N-60°N especially over the Eurasian continent. The spatial distribution of the total ice water path (IWP) changes resembles that of the total CWP changes (Fig. 10g-i).

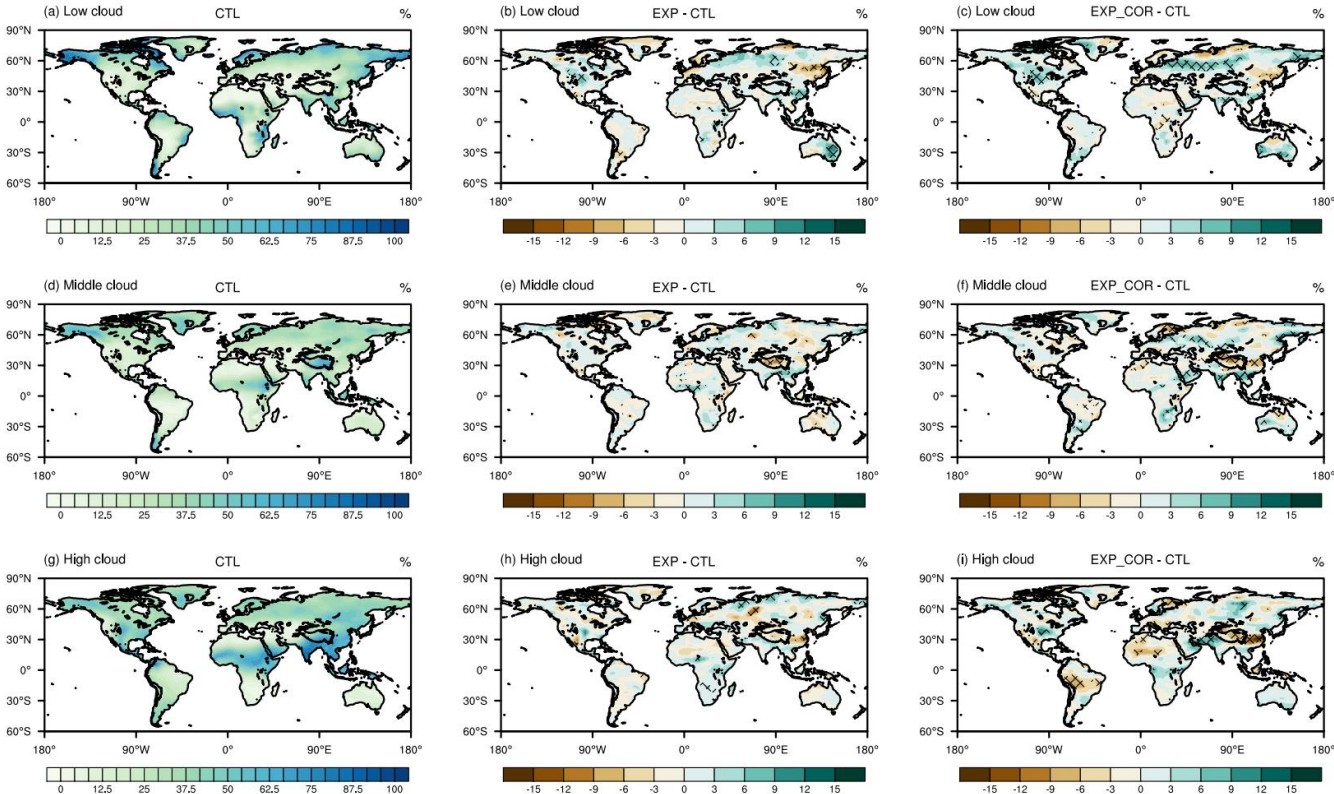

**Figure 9. Spatial distributions of the CTL run (left), and the differences in low clouds (the first row), middle clouds (the second row),**
**and high clouds (the third row) between EXP and CTL (middle) and between EXP_COR and CTL (right). The crossed areas are significant at the 95% level.**

Generally, the radiative effect of clouds is quantified by cloud radiative forcing (CRF) (the difference in the surface net flux between all sky and clear sky conditions). It includes shortwave cloud forcing (SWCF) and longwave cloud forcing (LWCF). Realistic simulation of the CRF is another important measure of the performance of climate models (Sun et al., 2016). The

SWCF is negative, and a smaller value indicates a stronger reflection of the solar shortwave radiation. Fig. 10b demonstrates that in the EXP run, SWCF is weakened over northeastern China, the TP, and southeastern Russia and is enhanced over

southern China, central Africa, the central US, southeastern Australia and along the Bay of Bengal compared with the CTL run. In the EXP_COR run, the reductions are mainly located in northern China, the TP, and southeastern Russia, while the enhancements are over the central US, over the Eurasian continent along 60°N, over southern China and along the Bay of Bengal. The increased SWCF originates from the increased cloud water (Fig. 10f) and low clouds (Fig. 9c) reflecting more solar shortwave radiation, while the decreased SWCF is due to the reduction of cloud water content and cloud fraction. The LWCF is positive, and a larger value means a stronger warming effect on the land surface. The LWCF increases over southern China and decreases over northern China (figure not shown). The distribution of the net CRF (figure not shown) resembles that of the SWCF, which implies that the SWCF is dominant in the CRF variations.

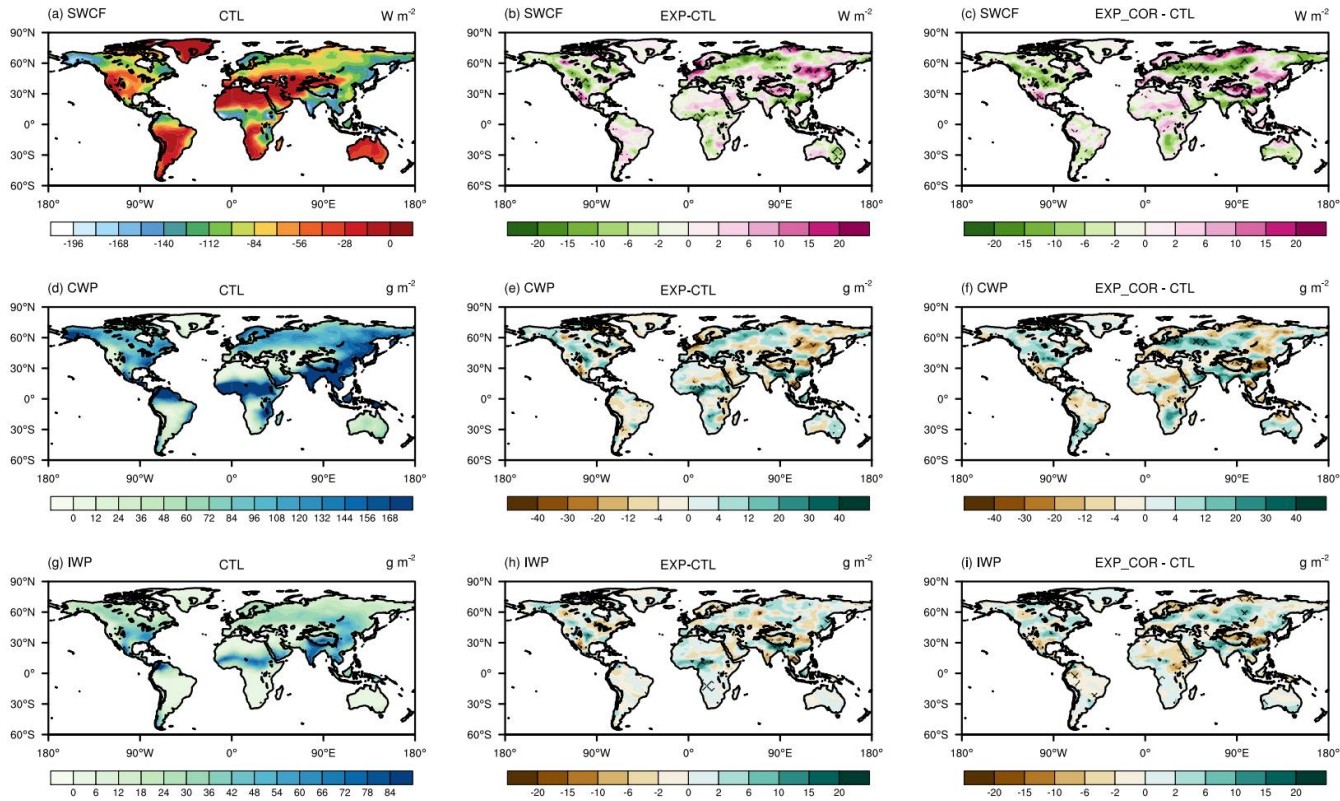

**Figure 10. Same as Figure 9 but for (a - c) shortwave cloud radiative forcing (units: W m⁻²), (d - f) total cloud water path (units: g m⁻²), and (h - k) ice water path (units: g m⁻²).**

The simulation of the net surface shortwave flux is demonstrated in Fig. 11. Globally, the averaged RMSE and COR are similar to each other in the three simulations. In the EXP_COR run, the underestimation over northern China and the TP in both the CTL and EXP runs is alleviated, although it slightly degrades the simulated shortwave flux over southern China. The negative biases over southeastern Russia in EXP_COR are also larger than those in EXP. The changes in the net surface shortwave flux (Fig. 11e&f) are very consistent with those in SWCF (Fig. 10b&c) implying that the net surface radiation fluxes are mainly dominated by the shortwave radiation reflected by the adjustment of clouds as a result of the altered PBL heating rates and the

associated local convection. The simulated patterns of the net surface shortwave and longwave fluxes (upward positive) are essentially consistent (figure not shown). The more (less) shortwave radiation is received by the surface, the more (less) heat it directly obtains, and the more (less) sensible heat flux it emits to warm (cool) the atmosphere.

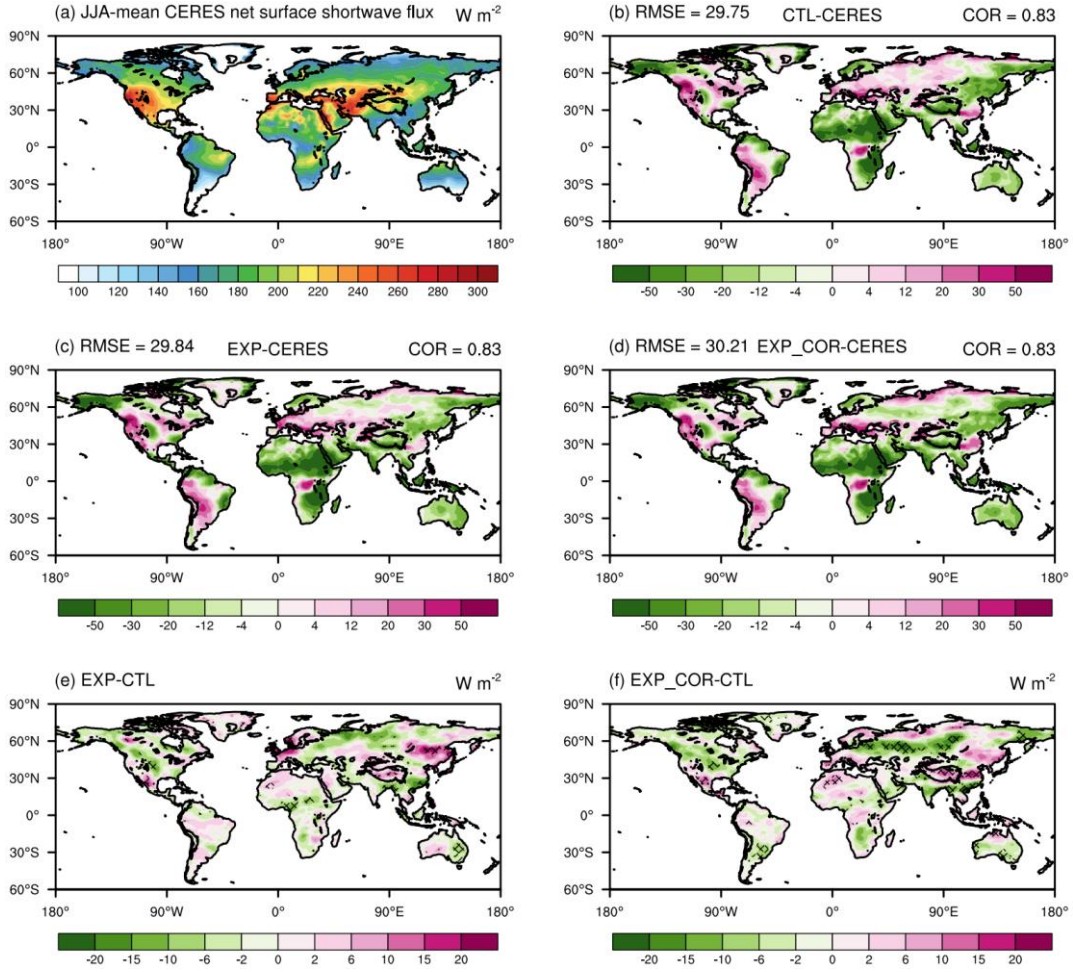

**Figure 11. Spatial distributions of the JJA-mean net surface shortwave flux in (a) CERES-EBAF (downward positive), the biases of (b) CTL, (c) EXP, and (d) EXP_COR with respect to CERES-EBAF, and the differences between (e) EXP and CTL and between (f) EXP_COR and CTL. The crossed areas are significant at the 95% level. The averaged spatial COR and RMSE for the three simulations are given in (b - d).**

In response to the adjustment of the surface energy budget, the global distributions of JJA mean 2 m air temperature from CRU and the difference between the observations and the three experiments are shown in Fig. 12. Overall, the three simulations have comparable CORs and RMSEs globally. Compared with the CTL run, the EXP run alleviates the overestimations in the middle and high latitudes, although the performance over central Africa and northern South America is slightly degraded (Fig. 12b, c&e). The positive biases over southern China are reduced in the EXP run. However, over northeastern China and southeastern Russia, the improvements are not significant. In the EXP_COR run, the overestimations over the central US and

the Eurasian continent are alleviated, while the negative biases over central Africa and the positive biases over southern South America are worsened (Fig. 12b, d&f). The simulated 2 m air temperature over northern China, and the TP is increased reintroducing some positive biases. In short, in the EXP_COR run, the decreased net surface shortwave flux associated with the increases of low clouds and cloud water content over southern China, the central US, over the Eurasian continent along 60°N, and along the Bay of Bengal might contribute to the cooling there, while the warming on the TP and northern China is attributed to the increased net surface shortwave flux associated with the decreased cloud fraction and cloud water content (Figs. 9 - 11).

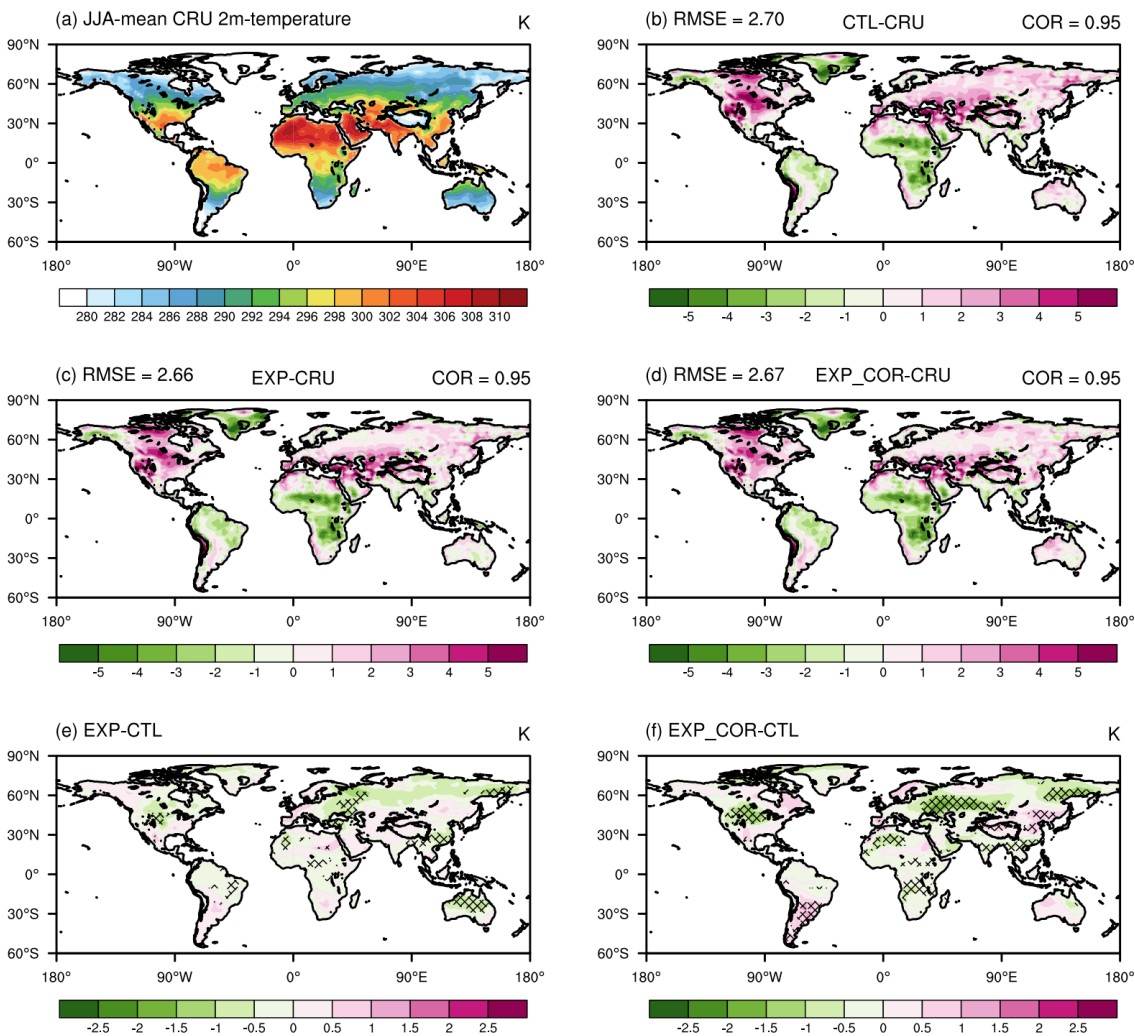

**Figure 12. Spatial distributions of the JJA-mean 2 m temperature in (a) the CRU, the biases of (b) CTL, (c) EXP, and (d) EXP_COR with respect to CRU, and the differences between (e) EXP and CTL and between (f) EXP_COR and CTL. The crossed areas are significant at the 95% level. The averaged spatial COR and RMSE for the three simulations are given in (b - d).**

## 3.3 Mean States

The analyses presented above demonstrate that the introduction of the subgrid heat flux schemes (EXP and EXP_COR), compared to the default model, improves the simulations of summer precipitation in Asia. The improvements and degradations in simulated surface heat fluxes, cloud properties and 2 m air temperature in boreal summer at the global scale are also discussed. The precipitation improvements over eastern China are mainly from the consideration of subgrid variations in surface heat fluxes (i.e., the EXP run where the sampled subgrid sensible and latent heat fluxes are stochastically paired with each other), while the improved precipitation simulations on the southern and eastern margins of the TP are attributed to the further inclusion of the partitioning of the subgrid surface heat fluxes (the EXP_COR run). A thorough evaluation of the global annual and seasonal means of those variables is necessary because from the perspective of climate model development, the incorporation of a new parameterization scheme to improve some aspects should not cause the degradation of other aspects (Wang et al., 2021b). As presented in Table 1 (global distributions shown in Figs. S3-9), overall, the simulation statistics of the EXP and EXP_COR runs are comparable to those of the CTL run, although slightly different in some seasons. When focusing on East Asia (Table S1), the new schemes outperform the default scheme in terms of COR and RMSE, implying the necessity and importance of parameterizing the subgrid land surface heat fluxes to the atmosphere in GCMs in regions with complex terrain (e.g., the TP) and multiple surface types (e.g., eastern China).

**Table 1. The COR and RMSE values in the CTL, EXP and EXP_COR runs. MAM is for March-April-May, JJA for June-July-August, SON for September-October-November, and DJF for December-January-February. The best performance among the three experiments is highlighted in bold.**

| Variables | Period | COR | | | RMSE | | |
|---|---|---|---|---|---|---|---|
| | | CTL | EXP | EXP_COR | CTL | EXP | EXP_COR |
| Precipitation | MAM | **0.82** | **0.82** | 0.81 | **1.55** | **1.55** | 1.61 |
| | JJA | 0.78 | **0.80** | 0.79 | 2.11 | **2.03** | 2.04 |
| | SON | 0.85 | 0.85 | 0.85 | 1.53 | **1.52** | 1.53 |
| | DJF | **0.85** | 0.84 | 0.84 | **1.62** | 1.65 | 1.73 |
| | Annual | 0.86 | 0.86 | 0.86 | 1.29 | **1.27** | 1.30 |
| 2 m Temperature | MAM | 0.98 | 0.98 | 0.98 | 2.57 | 2.50 | **2.49** |
| | JJA | 0.95 | 0.95 | 0.95 | 2.70 | **2.66** | 2.67 |
| | SON | 0.98 | 0.98 | 0.98 | 2.64 | 19.94 | **2.61** |
| | DJF | 0.99 | 0.99 | 0.99 | 4.01 | **3.76** | 3.80 |
| | Annual | 0.98 | 0.98 | 0.98 | 2.50 | 5.86 | **2.42** |
| Sensible Heat Flux | MAM | **0.67** | 0.65 | 0.65 | **34.08** | 34.73 | 34.43 |
| | JJA | 0.55 | **0.56** | **0.56** | 30.67 | **30.57** | 30.89 |
| | SON | 0.86 | 0.86 | 0.86 | **23.40** | 25.79 | 23.92 |
| | DJF | **0.88** | 0.87 | 0.87 | **23.71** | 24.42 | 24.42 |
| | Annual | **0.74** | 0.73 | 0.73 | **22.71** | 23.72 | 23.28 |
| Latent Heat Flux | MAM | **0.89** | 0.88 | 0.88 | **15.84** | 16.37 | 16.23 |
| | JJA | **0.82** | **0.82** | 0.81 | 24.24 | **23.18** | 23.40 |
| | SON | 0.88 | 0.88 | 0.88 | 17.34 | 17.57 | **17.33** |

| | | | | | | |
|---|---|---|---|---|---|---|
| | DJF | **0.92** | 0.91 | 0.92 | **15.99** | 16.93 | 16.44 |
| | Annual | 0.90 | 0.90 | 0.90 | **13.92** | 14.17 | 14.15 |
| Net Surface Shortwave Flux | MAM | **0.92** | 0.91 | 0.91 | **21.89** | 23.20 | 23.47 |
| | JJA | 0.83 | 0.83 | 0.83 | **29.75** | 29.84 | 30.21 |
| | SON | 0.96 | 0.96 | 0.96 | **20.35** | 26.06 | 21.10 |
| | DJF | 0.96 | 0.96 | **0.97** | **24.28** | 24.51 | 24.32 |
| | Annual | 0.93 | 0.93 | 0.93 | **19.35** | 21.04 | 20.05 |

The zonal means of temperature and specific humidity from the European Centre for Medium-Range Weather Forecasts (ECMWF) ERA-Interim reanalysis dataset and the model biases are shown in Fig. 13. In the CTL run, the temperature is overestimated at lower levels in the tropics and midlatitude regions in the SH, whereas at other latitudes and levels, it is underestimated (Fig. 13a). The EXP run reverses the positive biases back to negative biases with an excessive reduction at lower levels, and the negative biases in other regions are further exacerbated (Figs. 13b&S10b). In contrast, the biases in the EXP_COR run are comparable to those in the CTL run (Fig. 13a&c). The low-latitude overestimations in the lower troposphere and the high-latitude underestimations across the troposphere are alleviated to some extent (Fig. S10c). In the simulation of specific humidity, compared to the observations, the main positive biases occur in the low latitude and midlatitude regions below 400 hPa. For the midlatitude region, there are negative biases at lower levels (Fig. 13d). Generally, the performance among the three simulations is similar to each other (Figs. 13d-f). In the EXP run, the biases are alleviated (Fig. S10e). The values of the EXP_COR run are comparable to those of the CTL run, and their differences are minor and negligible from the perspective of the annual zonal averages (Fig. S8f). In summary, the performance of the mean state simulations does not change significantly when using the two modified schemes (the EXP and EXP_COR runs) in terms of the variables discussed above.

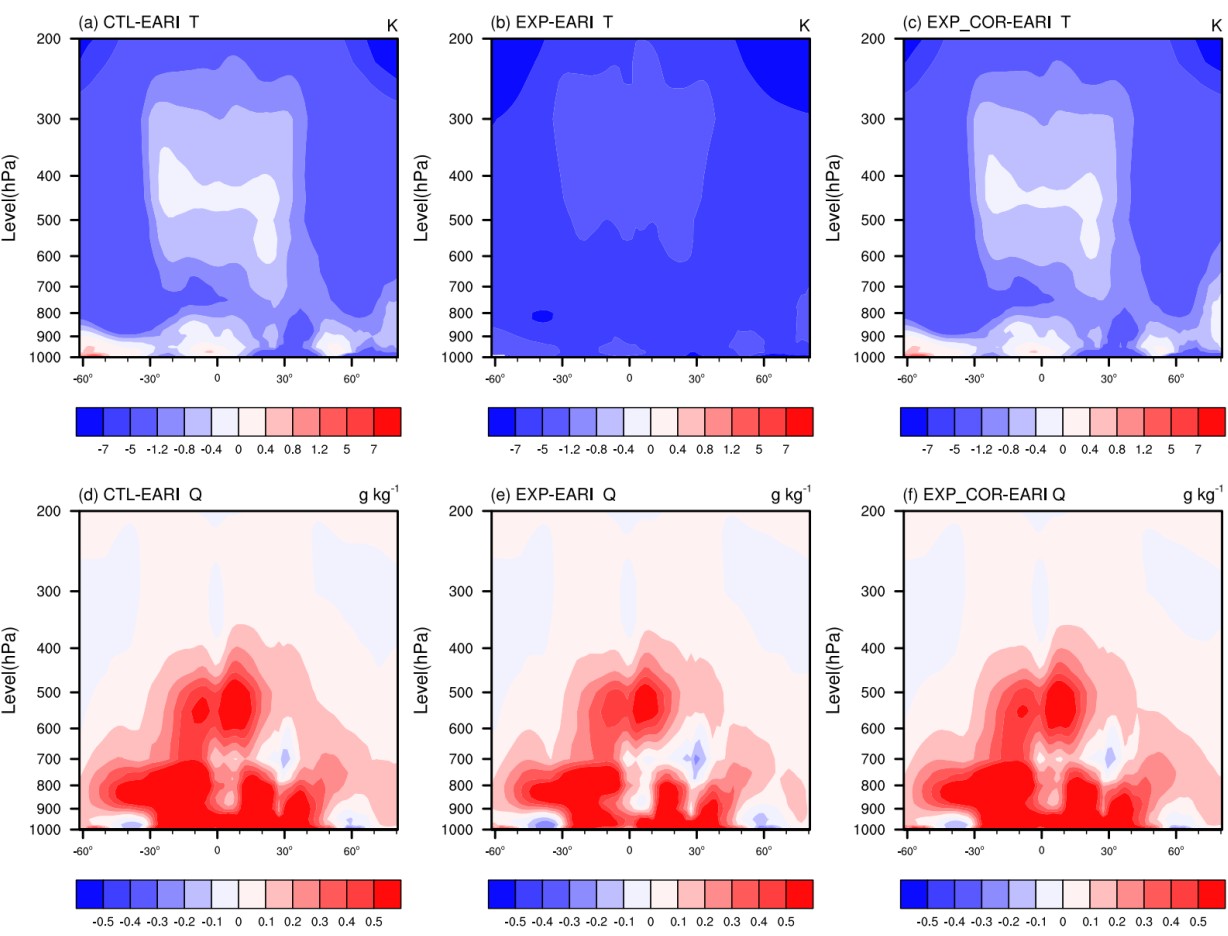

**Figure 13. Annual and zonal mean cross-sections of the (a–c) temperature and (d–f) specific humidity differences for (a&d) CTL-ERAI, (b&e) EXP- ERAI , and (c&f) EXP_COR- ERAI.**

## 4 Discussion

Despite the uncertainties in the observations, the overestimated rainfall on the southern and eastern margins of the Tibetan Plateau in the GCMs is widely acknowledged when comparing multiple observations (Mehran et al., 2014; Yu et al., 2015). The uncertainties for the evaluations of other modeled variables are discussed below. The CERES-EBAF datasets provide long-term global Earth radiation budget records from the surface to the top of the atmosphere (TOA) together with the associated cloud and aerosol properties. Extensive validation has been conducted for both TOA and surface radiation in CERES-EBAF using TOA consistency tests and direct comparisons of surface fluxes with ground-based measurements over both land and ocean (Loeb et al., 2007; Loeb et al., 2012). Although some weaknesses are noted (e.g., LW cloud radiative effects at the surface on the TP are overestimated due to poor sampling of clear sky scenes during the night), they are widely used for climate model evaluations (Loeb et al., 2018; Hinkelman et al., 2019), and this flaw does not affect the conclusions

in this study. As for surface sensible and latent heat fluxes, there are few observations covering the whole TP. Instead, among various reanalysis datasets, GLDAS has been evaluated and investigated extensively (Novick et al., 2018; Sun et al., 2018; Laloyaux et al., 2016). For instance, Jiménez et al. (2011) conducted a global intercomparison of monthly mean land surface heat flux products, including space-based observations and reanalyses including GLDAS. They demonstrated that the spatial distributions related to the major climatic regimes and geographical features are well reproduced by GLDAS. With comprehensive validations, the GLDAS product has been widely used in evaluating model-based studies (Saha et al., 2014; Xia et al., 2019) such as water resource management (Zaitchik et al., 2010), and drought monitoring and prediction (Hao et al., 2016). The CRU gridded dataset for 2 m air temperature has undergone a series of technical validations, such as quality control of input data, comparisons between versions and with alternative datasets, and cross-validation of the interpolated anomalies (Osborn et al., 2017; Harris et al., 2020).

In addition to subgrid variation and partitioning of surface heat fluxes, there are other factors that can impact the precipitation simulation on the TP. For instance, subgrid topographic effects have large effects on latent heat and sensible heat fluxes. It is found that parameterizing them in GCMs influences the simulated surface energy balance and boundary conditions, as well as precipitation on the TP (Lee et al., 2019; Hao et al., 2021, 2022). Alternatively, the accurate representation of land cover types and soil properties is vital to the realistic simulation of surface radiative and heat fluxes and thus TP rainfall (Liu et al., 2021; Yue et al., 2021).

With 208 CPU cores in total for each simulation, the total run time per step (~0.50 sec) in the EXP_COR run is almost twice that in the CTL run (~0.26 sec) as a result of calling the PBL and convective parameterizations 16 times and the resulting extra communication cost (Table S2). However, compared with the four-mode version of the Modal Aerosol Module (MAM4) updated from MAM3 and the Cloud Layers Unified by Binormals (CLUBB) scheme instead of the CAM5 boundary layer turbulence, shallow convection, and cloud macrophysics schemes in CAM6, the computational cost here is much smaller and thus acceptable. Given the heavy computational cost of CLUBB, this could be challenging for computational efficiency if using this scheme in CESM2 (CESM version 2). Therefore, further improvements are needed. For example, according to the number of PFTs in each grid cell, the number of multiple calls (up to 16) of the CLUBB can be varied in different grid cells. Alternatively, do this only when the number of PFTs is larger than a threshold. In the meantime, parallel optimization should be applied to multiple calls.

The GCM used to test the schemes is CESM1.2, in which the land model is CLM4. Similar to CLM4, CLM5 (CLM version 5) in CESM2 and other land surface models in the GCMs use the PFT structure as well. Additionally, the parameterization of subgrid heat fluxes proposed in this study is not dependent on the specific parameterizations of the PBL and convection processes. Therefore, it is conveniently applied to other GCMs.

## 5 Conclusions

In this study, a parameterization of the subgrid variations and partitioning of the land surface heat fluxes to the atmosphere was developed and implemented in the NCAR CESM1.2. The modification to the Sun et al. (2021) scheme is based on the fact that energy redistribution with complex climate impacts between the land surface and the PBL plays an essential part in the global and regional energy cycles (Liu et al., 2014; Chakraborty and Lee, 2019; Wei et al., 2021). Three experiments were conducted to evaluate the updated scheme (CTL, EXP, and EXP_COR). The precipitation improvements over eastern China derived using the original scheme (EXP) still exist in the new scheme (EXP_COR), although slight dry biases are reintroduced over southern China. In addition, the stubborn overestimations of precipitation on the southern and eastern margins of the TP are alleviated.

The causes are briefly summarized in Fig. 14a. The subgrid variations of the land surface heat fluxes increase (decrease) PBL heating over southern (northern) China. With the further introduction of the partitioning of subgrid surface heat fluxes, the increase (decrease) in PBL heating over southern (northern) China is elevated, thus destabilizing (stabilizing) the lower atmosphere. As a result, local convection is promoted (suppressed) over southern (northern) China. The changes of convective precipitation dominate the changes of total precipitation over eastern China and the eastern margin of the TP. The altered large-scale circulation associated with the easterly anomaly along 25° N-30° N partly blocks moisture transport from the ocean in the south to the southern margin of the TP. Accordingly, the decrease of large-scale precipitation is responsible for the reduced precipitation there.

The links among clouds, net surface shortwave flux and 2 m air temperature over eastern China are shown in Fig. 14b. As PBL heating decreases in northern China, the lower atmosphere stabilizes and local convection is suppressed. Accordingly, middle and high clouds, and the associated CWP decrease (Figs. 9&10). Thus, SWCF decreases over northern China, which increases the net surface shortwave flux. As the surface gains more energy, the near-surface air temperature warms. In contrast, southern China features the opposite changes in the storyline.

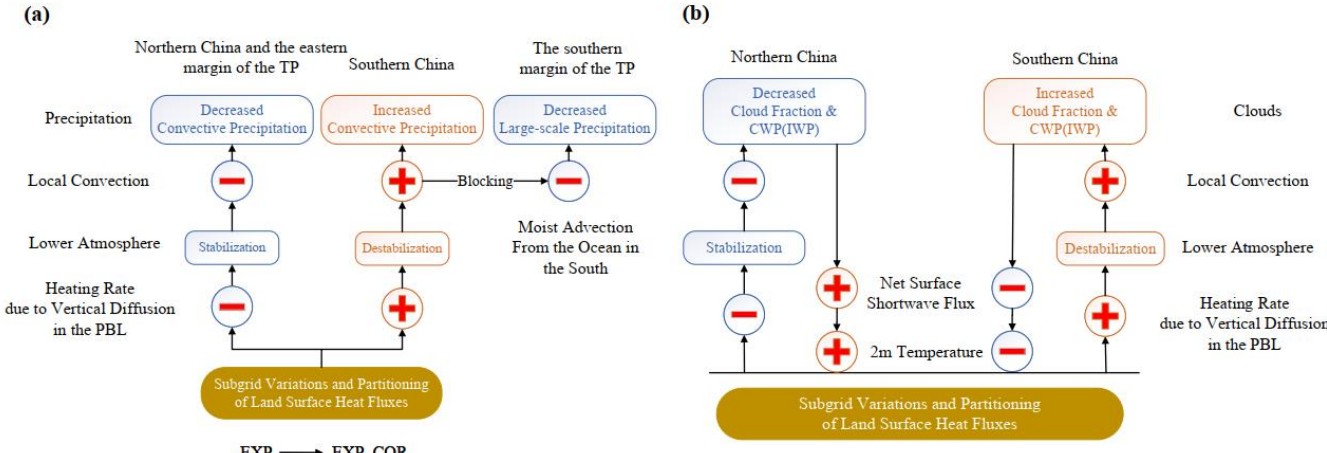

**Figure 14. Schematic diagram summarizing the climate impacts of parameterizing subgrid variations and partitioning of land surface heat fluxes to the atmosphere.**

The Sun et al. (2021) scheme offers a novel method of parametrizing the subgrid heterogeneity of surface heat fluxes to the atmosphere in GCMs. As a further modification, the significance of the correlation coefficients between the subgrid-scale sensible and latent heat fluxes is considered for a more realistic interpretation of the energy exchange processes. The findings of these two studies highlight the importance of the energy variation and redistribution between the land surface and the lower atmosphere at the subgrid scale.

### Code and data availability

The CESM1.2.1-CAM5.3 source code can be downloaded through the CESM official website https://www.cesm.ucar.edu/models/cesm1.2/cesm/doc/usersguide/x290.html#download_ccsm_code. The modified CESM code as well as the CAM5 output for all simulations in the study are provided in an open repository Zenodo (https://zenodo.org/record/6606418#.YpiHWKhBw2w). The TRMM data are available from https://gpm.nasa.gov/data/directory. The MERRA-2 data files are available from https://disc.gsfc.nasa.gov/datasets/M2IMNPASM_5.12.4/summary?keywords=M2IMNPASM_5.12.4%20instM_3d_asm_Np and https://disc.gsfc.nasa.gov/datasets/M2TMNXFLX_5.12.4/summary?keywords=M2TMNXFLX_5.12.4%20tavgM_2d_flx_Nx. The CERES EBAF data are available from https://climatedataguide.ucar.edu/climate-data/ceres-ebaf-clouds-and-earths-radiant-energy-systems-ceres-energy-balanced-and-filled. The GLDAS-2.1 data are available from https://disc.gsfc.nasa.gov/datasets/GLDAS_NOAH10_M_2.1/summary?keywords=GLDAS. The CRU data are available from https://crudata.uea.ac.uk/cru/data/hrg/?_ga=2.162163900.162961233.1636977076-620633058.1635581908.

### Author contribution

YW conceived the idea. WS developed the model code. WS and YH conducted the model simulations. MY and YW performed the analysis. MY and YW interpreted the results and wrote the paper. MY, YH and YW revised the manuscript. All authors participated in the discussion of the paper.

### Competing interests

The authors declare that they have no conflicts of interest.

**Disclaimer**

Publisher's note: Copernicus Publications remains neutral with regard to jurisdictional claims in published maps and institutional affiliations.

**Acknowledgments**

YW is supported by the National Natural Science Foundation of China Grant 41975126 and the National Key Research and
Development Program of China Grant 2017YFA0604000. We thank the two reviewers for the comments which significantly improved the quality of the paper.

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
