# Peer review of "Climate Impacts of Parameterizing Subgrid Variation and Partitioning of Land Surface Heat Fluxes to the Atmosphere with the NCAR CESM1.2"

_Geoscientific Model Development, 2022_

## Referee Comment (RC1)

**Yin et al (2022): "Climate Impacts of Parameterizing Subgrid Variation and Partitioning of Land Surface Heat Fluxes to the Atmosphere with the NCAR CESM1.2"**

**Overview**

*Yin et al.* (2022) builds on work presented in *Sun et al.* (2021), which introduced a new parameterization to capture the impacts of subgrid surface fluxes by randomly sampling from distributions of sensible and latent heat fluxes within a gridcell and using those samples to drive unique realizations of the PBL and convection schemes in CESM1.2. *Yin et al.* expand on that to include a condition that the sensible and latent heat fluxes being sampled are either positively or negatively correlated with one another to better capture their relationship and land-atmosphere interactions. The method is novel and holds promise for inclusion in GCMS, though I believe major revisions are needed before this work is published. My main concern is that while model improvements stemming from EXP_COR are strongly emphasized, regions where performance has been degraded are rarely discussed. It is not yet clear that this version of the parameterization represents a strong improvement over the original EXP run. The discussion around each of the figures requires major revision as a result. It would also be beneficial to highlight the model performance globally with more detail.

**Major comments**
- Section 2.1: Since the parameterization introduced in *Sun et al.* 2021 is still so new and not yet widely known/implemented, further explanation and clarification should be included here. More details of the formulation of the normal distribution (i.e., their Eq. 3) would be helpful at minimum.
- Line 126: The PBL and convective parameterizations are called 16 times per time step to fully sample these subgrid fluxes; what's the impact of that on model efficiency (run time vs. throughput)? Would this be cost prohibitive to implement in the current version of CAM (CAM5) for example, due to its use of CLUBB? It's suggested later that this parameterization could be easily added to CESM and other GCMS, but this information is key to that statement.
- Lines 148-150: I appreciate the need to limit the scope of this study but given that a portion of the motivation for this work was a more through exploration of other variables (2m temperature, surface fluxes, clouds, etc.) and that this is a modified parameterization relative to *Sun et al.* – it would be useful to first confirm that this is still a critical region/season where the new scheme produces a sizable impact.
  - It's noted that global mean values are assessed in Section 3.3, but there are no spatial maps either annually or seasonally, which could strengthen the study.
- Lines 154-155: "It is encouraging that after taking the subgrid energy partitioning into account, the longstanding biases are efficiently mitigated in the EXP_COR run." This is not readily apparent in the bias plots (Fig 2f vs. 2d). Though I see that relative to the CTL

(Fig 2e), there is indeed a small decrease in rain rate along the southern edge of the TP, the bias is still over 5 mm/day. It also appears that a dry bias has been re-introduced over Eastern China in EXP_COR that had been mitigated in EXP. A more thorough discussion of where EXP_COR does *not* improve precipitation is warranted.

- Figure 3: My understanding is that the sum of panels 3a and 3c should match Fig. 2c, and that 3b+3d should equal 2e – is this correct? If so, I wonder if there is a slight plotting mismatch; in Fig 2c, changes in rainfall rarely extend past the TP border, and the small region that does is on the order of 0.5-1 mm/day. But in Fig 3a and 3c, there are precipitation increases that extend along much of the southeastern TP border and into the TP itself, which can be on the order of 1-2 mm/day.

- Lines 200-201: "In contrast, in the EXP_COR run (Fig. 5d), the SLP simulation is corrected…" This does not appear to be the case. The decrease in SLP on the TP in the EXP_COR vs. the CTL case is less than 1 hPa, but the bias in the CTL relative to MERRA2 is more than 10 hPa; the shift is quite small and does not correct the existing bias (though it does reduce it slightly).

- Lines 217-218: "…the underestimations on the southern and eastern margins of the TP in the CTL runs are remarkably improved (Fig. 6f)." When looking for model improvements, it would make more sense to focus on the comparison against observations rather than the control, which would be Fig. 6d. In that light, there is again certainly some improvement in the model bias, but the scale of that change is perhaps minor: on the order of what looks like 2-4 W/m2 when the bias is more than 20 W/m2. The language used in the text suggests a stronger change than is visible.

- Lines 220-221: "The sensible heat flux changes in the EXP_COR run are more significant than those in the EXP run (Fig. 7f), especially over northern China and the southern and eastern margins of the TP, resulting in better agreement with observations." The change over northern China seems to be stronger in the EXP than the EXP_COR simulation, while the increase in SHFLX over eastern China seems to have actually *increased* the bias there relative to both the CTL and EXP cases. Focusing in on the southern edge of the TP may miss out on the bigger regional picture in this case.

- Lines 251-252: "In the EXP_COR run, in addition to the already existing significant improvements in the EXP run…" While it's stated that the EXP run alleviates the overestimation of surface shortwave flux in southern China, it is important to note that the EXP_COR run reduces that improvement and even worsens the bias over eastern China (Fig. 2d). It also seems that the biases in EXP_COR are larger than in EXP over north eastern China/southern Russia, despite general improvements over the TP.

- Figure 12: Unfortunately this diagram is incredibly hard to read. The legend and key need to use a much larger font size to be readable, and even within the plot itself, distinguishing shapes/numbers from each other is challenging. A table may be easier to read, but at minimum the Taylor diagram needs to be heavily revised.
  - Section 3.3: Differences in the mean state are quite hard to distinguish given the readability issues of Fig. 12.

- Lines 328-330: "In summary, the performance of the mean state simulations does not change significantly when using the modified scheme (EXP_COR), indicating that the

subgrid parameterization scheme can be incorporated into the GCMs without heavy retuning." This may be a bit overstated. It's important to acknowledge that the current analysis is dependent only on annual, global/zonal means, and only a handful of variables. Spatial/seasonal maps could strengthen the arguments made.

- Lines 382-383: "The mean states did not change much after the introduction of the new parameterization scheme, and thus, the new scheme can be implemented in the CESM without heavy retuning." In addition to the concerns raised above, it's important to note that these experiments are using an outdated version of CESM. In order to be incorporated into the model, these results would need to be reassessed using CESM2; some explanation of why CESM1.2 vs. CESM2 was used could be helpful in this case, and/or whether there are potential challenges to using this within CESM2 vs. CESM1.2.

**Minor comments**
- Lines 34-36: "This results in most of the precipitation simulation errors in GCMs, such as…" Are the authors saying that the use of grid mean rather than subgrid surface fluxes is responsible for the majority of precipitation biases in GCMs? That may be an exaggeration; the studies cited in the rest of the second sentence seem to suggest a range of other ways to mitigate precipitation biases, in fact. Suggest rephrasing for clarity.
- Line 36: Suggest additional citations on the bias of the rainfall intensity spectrum, as this is a frequently recognized bias that was recognized well before 2021 – the historical context is useful.
- Line 42: "…changes in vegetation density have been found to favor the release of latent heat…" Please be specific in what direction the changes are occurring; does an increase or a decrease in vegetation density link to favored LHFLX?
- Fig 1: Does this spatial pattern of correlation coefficients change with season? It would likely be more useful to the reader to consider DJF/JJA means at least to understand the importance of the correlation coefficient being a time-varying quantity rather than one that is static in time.
- Line 123: Please clarify the time scale at which the correlation coefficient r varies; is this computed at each time step?
- Section 2.3: Are the seasonal averages based on monthly means or higher temporal resolution? Daily data may yield a more realistic picture of model performance, even though it is averaged up to the seasonal level in analysis.
- Figure 2: Since the main region of interest in the TP, it would be helpful to also give the RMSE and correlation for that specific region (as it seems may be the norm in other figures of this paper).
- Figure 3 (and other figures): Suggest adding labels that denote "EXP – CTL" and "EXP_COR – CTL" to each subpanel, as in Fig 2.
- Lines 196-200: This section seeks to draw comparisons to the EXP and EXP_COR simulations, but there are no EXP-related plots in Fig. 5; please reproduce figures from *Sun et al*. that are key to this analysis within the current paper.

- Figure 5: In addition to including plots of the EXP case, it would be helpful to also include the raw EXP_COR analysis and (space permitting for this) the difference from MERRA2 observations and not just the CLT case. This would make the analysis points that are made much more convincing and clear.
- Line 211: As noted above, it does not appear that the atmospheric circulation is "corrected" in this case, so I suggest softening the language.
- Figure 6 (and others): Please clarify if the RMSE and COR values are *just* for the inset region over the TP? It would be helpful in general to have these values for both the full domain *and* the TP region itself.
  - It would also result in a cleaner plot to simply place a rectangle around the TP region where those values are calculated (perhaps in panel a) rather than an additional inset plot since the TP region is not any larger than in the main subpanel.
- Line 269: "It should be noted that there is a significant negative band along 60˚N." More explanation should be included if this is to be noted – why is this band present in EXP_COR but not EXP, does this suggest better/worse agreement with obs, etc.
- Fig 10: In this figure and similar ones that follow, it would be useful to show the raw CTL case values as well so that the reader can determine how large each change is relative to the baseline. It would also again be useful to include "EXP-CTL" and "EXP_COR-CTL" in the plot titles.
- Fig. 13: It would be helpful to show the difference in bias (relative to obs) for each case, not just the difference in each case relative to CTL. That would make the statements in section 3.3 more clearly supported.
- Lines 369-372: The precipitation improvements from EXP are *mostly* present in EXP_COR, but Fig. 2f suggests a reintroduction of a dry bias over eastern China that had been removed in EXP, which should be noted. The overestimates in precipitation along the southern/eastern borders of the TP are also still present and of a large amplitude, so with the current colorbar it is hard to deduce that those biases are "significantly alleviated."

---

## Referee Comment (RC2)

**Reviewer's report for gmd-2022-114**

**General comments**

The authors briefly summarized the subgrid surface heat flux scheme initially proposed by Sun et al. (2021) and specifically introduced the modified version. These two stochastic sampling models actually introduce the uncertainties into the land-atmosphere coupling process other than, as the authors claimed, represent the realistic surface heat flux partitioning. How does the model deal with the surface energy balance closure in accompany with the imposed heat flux partitioning? On the assumption of normal distribution of the surface heat fluxes, how many land grid cells are subject to that normal distribution if the purpose of this study is to parameterize the realistic surface heat fluxes? And to what extent the other "abnormal" grid cells can play roles in impacting the simulated climate? I understand that it may be difficult to address these points at one time, but it will benefit the readers if the authors can provide some advantages and disadvantages of Sun's approach and the modified approach in the text.

As the manuscript reads now, it seems like the authors touched on a wide range of analysis of the atmospheric variables only briefly without really figuring out their scientific connections. Instead, the authors may consider restructuring the manuscript and dig deeper into aspects that are truly linked with the incorporation of the subgrid-scale treatment of surface heat fluxes. For example, the authors stated that the simulated large-scale circulation is significantly altered by the modified surface heat fluxes which further improves the simulated precipitation on the southern and eastern margins of TP. If so, is it possible to demonstrate the relationship between the changes in surface heat fluxes (Figs. 6&7) and the affected SLP field in Fig. 5? Also, it is very interesting to see the cloud feedbacks in EXP or EXP_COR, and that is supposed to be induced by the surface energy changes in this study. But in the context, the changes in cloud properties are considered the reason why surface heat fluxes are influenced.

**Specific comments**

L75: Suggest changing the title to "Materials and methods" or "Methodology"

L82: "all of the PFTs in the grid" -> "the corresponding grid"

L88: "selected as N pairs" -> "paired with each other"

L97: "normal distributions" see the general comment

L152: "Sun et al. improved … (Fig. 2c)" Should it be Fig. 2d or Fig. 2c? Fig. 2c is comparing Sun et al. results with the control case.

L157: Given that the RMSE of EXP_COR is very close to EXP, is it sensitive to the average region the authors defined? Or, why do the authors choose 0-50°N, 0-180°E as the study area for the statistics?

L198-L200: If the authors can include the Fig.5b from Sun et al. (2021) into Fig. 5 in this manuscript, it may be easier for readers to understand the context on explaining model's overestimated precipitation in the southern TP.

Figure 5: Does it show the moisture transport vector or wind speed? The unit as depicted in Fig. 5 is m s-1.

Figures 6-11: The inserted boxes do not zoom in much or indicate more useful information. They may be removed.

L221: Remove "resulting in better agreement with the observations"

L303-L305: This sentence is a little confusing. What is the difference between "subgrid variations in surface heat fluxes" and "their subgrid partitioning"? It might be good to provide an explanation.

Figure 12: Could the authors make this blurry plotting of a higher quality?

---

## Author Comment (AC1)

**Reply to the comments by Referee #1**

We thank the reviewer for the comments and suggestions to improve our manuscript. Below is our point-by-point response to these comments. The reviewer's comments are in italics, our responses are in normal font, and manuscript revisions are in blue.

*Yin et al. (2022) builds on work presented in Sun et al. (2021), which introduced a new parameterization to capture the impacts of subgrid surface fluxes by randomly sampling from distributions of sensible and latent heat fluxes within a grid cell and using those samples to drive unique realizations of the PBL and convection schemes in CESM1.2. Yin et al. expand on that to include a condition that the sensible and latent heat fluxes being sampled are either positively or negatively correlated with one another to better capture their relationship and land-atmosphere interactions. The method is novel and holds promise for inclusion in GCMS, though I believe major revisions are needed before this work is published.*

**Reply**: We appreciate the reviewer's positive remarks.

*My main concern is that while model improvements stemming from EXP_COR are strongly emphasized, regions where performance has been degraded are rarely discussed. It is not yet clear that this version of the parameterization represents a strong improvement over the original EXP run. The discussion around each of the figures requires major revision as a result. It would also be beneficial to highlight the model performance globally with more detail.*

**Reply**: This study aims to present a thorough evaluation of the performance of two parametrizations on simulated climate variables rather than seek significant improvements to all the variables in the EXP_COR run compared to the original EXP run. One vital improvement we expected is to see whether the stubborn bias of the simulated precipitation over the southern and eastern margins of the TP in the EXP run (also seen in other GCM simulations) can be alleviated in the EXP_COR run. Following the suggestion, in the revised manuscript, we tuned down the tone by explicitly discussing both improvements and degradations in the EXP_COR run relative to the EXP run. The evaluations of all the variables at the global scale have been included. Please see the detailed response to the major and minor comments below.

***Major Comments***
***Comment 1***:
*Section 2.1: Since the parameterization introduced in Sun et al. 2021 is still so new and not yet widely known/implemented, further explanation and clarification should be included here. More details of the formulation of the normal distribution (i.e., their Eq. 3) would be helpful at minimum.*

**Reply**: Thanks for the valuable suggestion. We have included a more detailed introduction to the Sun et al. (2021) parameterization in Section 2.1 in the revision:

"This scheme established the truncated normal distributions of the subgrid sensible and latent heat fluxes independently within the grid cell at each time step. The probability density function (PDF) of subgrid sensible and latent heat flux in a given grid cell was calculated by

$$f(x|\bar{F}, \sigma, F_{min}, F_{max}) = \frac{\frac{1}{\sigma}\phi\left(\frac{x-\bar{F}}{\sigma}\right)}{\Psi\left(\frac{F_{max}-\bar{F}}{\sigma}\right)-\Psi\left(\frac{F_{min}-\bar{F}}{\sigma}\right)}, x \in [F_{min}, F_{max}] \qquad (1)$$

where $\bar{F}$ is the weighted average value of all subgrid heat fluxes, $\sigma$ is the standard deviation, $F_{min}$ and $F_{max}$ are the minima and maxima of the subgrid heat fluxes, respectively, and $\phi$ and $\Psi$ are the PDF and the cumulative distribution function (CDF) of the standard normal distribution, respectively." (Lines 86-92 in the revision)

***Comment 2***:
*Line 126: The PBL and convective parameterizations are called 16 times per time step to fully sample these subgrid fluxes; what's the impact of that on model efficiency (run time vs. throughput)? Would this be cost prohibitive to implement in the current version of CAM (CAM5) for example, due to its use of CLUBB? It's suggested later that this parameterization could be easily added to CESM and other GCMS, but this information is key to that statement.*

**Reply**: Thanks for this constructive suggestion. We have included Table R1 showing the run time per step and throughput for the CTL and EXP_COR runs in the supplementary Table S2 and discussed this in the discussion section in Lines 431-440 in the revision:

"With 208 CPU cores in total for each simulation, the total run time per step (~0.50 sec) in the EXP_COR run is almost twice that in the CTL run (~0.26 sec) as a result of calling the PBL and convective parameterizations 16 times and the resulting extra communication cost (Table S2). However, compared with the four-mode version of the Modal Aerosol Module (MAM4) updated from MAM3 and the Cloud Layers Unified by Binormals (CLUBB) scheme instead of the CAM5 boundary layer turbulence, shallow convection, and cloud macrophysics schemes in CAM6, the computational cost here is much smaller and thus acceptable. Given the heavy computational cost of CLUBB, this could be challenging for computational efficiency if using this scheme in CESM2 (CESM version 2). Therefore, further improvements are needed. For example, according to the number of PFTs in each grid cell, the number of multiple calls (up to 16) of the CLUBB can be varied in different grid cells. Alternatively, do this only when the number of PFTs is larger than a threshold. In the meantime, parallel optimization should be applied to multiple calls."

**Table R1.** Run time per step and throughput for the CTL and EXP_COR runs. Unit: second.

|  | CTL | EXP_COR |
|---|---|---|
| Total | 0.25642 | 0.48900 |
| Convection | 0.09949 | 0.19035 |
| PBL | 0.02286 | 0.09320 |
| CLM | 0.06643 | 0.06936 |
| Dynamic | 0.02696 | 0.02818 |
| Communication time | 0.01975 | 0.08248 |

***Comment 3***:
*Lines 148-150: I appreciate the need to limit the scope of this study but given that a portion of the motivation for this work was a more thorough exploration of other variables (2m temperature, surface fluxes, clouds, etc.) and that this is a modified parameterization relative to Sun et al. – it would be useful to first confirm that this is still a critical region/season where the new scheme produces a sizable impact. It's noted that global mean values are assessed in Section 3.3, but there are no spatial maps either annually or seasonally, which could strengthen the study.*

**Reply**: Thanks for the suggestion. The global distributions of the annual mean 2 m air temperatures, surface fluxes and clouds in the three simulations (Figs. R1-7) are included in the supplementary material as Figs. S3-9. The spatial correlation coefficients and root mean square error for annual and seasonal means (Table R2 here) are summarized in Table 1 in Sect. 3.3 instead of the Taylor diagram in the revision. The original figures regarding the differences among the three simulations in the main text have been updated with a global view (Figs. 3 and 7-12 in the revision). The related discussion has been added in Lines 379-380 in the revised manuscript:

"As presented in Table 1 (global distributions shown in Figs. S3-9), overall, the simulation statistics of the EXP and EXP_COR runs are comparable to those of the CTL run, although slightly different in some seasons."

[Figure]

**Figure R1.** Spatial distributions of the annual mean precipitation from (a) TRMM, (b) CTL, (c) EXP and (d) EXP_COR.

[Figure]

**Figure R2.** Spatial distributions of the annual mean 2 m temperature from (a) CRU, (b) CTL, (c) EXP and (d) EXP_COR.

[Figure]

**Figure R3.** Spatial distributions of the annual mean latent heat flux from (a) GLDAS, (b) CTL, (c) EXP and (d) EXP_COR.

[Figure]

**Figure R4.** Spatial distributions of the annual mean sensible heat flux from (a) GLDAS, (b) CTL, (c) EXP and (d) EXP_COR.

[Figure]

**Figure R5**. Spatial distributions of the annual mean net surface shortwave flux from (a) CERES, (b) CTL, (c) EXP and (d) EXP_COR.

[Figure]

**Figure R6**. Spatial distributions of the annual mean (a-c) low, (d-f) middle and (g-i) high cloud fraction for CTL (left), EXP (middle) and EXP_COR (right).

[Figure]

**Figure R7.** Spatial distributions of the annual mean (a-c) shortwave cloud radiative forcing (units: W m$^{-2}$), (d-f) total cloud water path (units: g m$^{-2}$), and (g-i) ice water path (units: g m$^{-2}$) for CTL (left), EXP (middle) and EXP_COR (right).

**Table R2.** The COR and RMSE values in the CTL, EXP and EXP_COR runs. MAM is for March-April-May, JJA for June-July-August, SON for September-October-November, and DJF for December-January-February. The best performance among the three experiments is highlighted in bold.

| Variables | Period | COR | | | RMSE | | |
|---|---|---|---|---|---|---|---|
| | | CTL | EXP | EXP_COR | CTL | EXP | EXP_COR |
| Precipitation | MAM | **0.82** | **0.82** | 0.81 | **1.55** | **1.55** | 1.61 |
| | JJA | 0.78 | **0.80** | 0.79 | 2.11 | **2.03** | 2.04 |
| | SON | 0.85 | 0.85 | 0.85 | 1.53 | **1.52** | 1.53 |
| | DJF | **0.85** | 0.84 | 0.84 | **1.62** | 1.65 | 1.73 |
| | Annual | 0.86 | 0.86 | 0.86 | 1.29 | **1.27** | 1.30 |
| 2 m Temperature | MAM | 0.98 | 0.98 | 0.98 | 2.57 | 2.50 | **2.49** |
| | JJA | 0.95 | 0.95 | 0.95 | 2.70 | **2.66** | 2.67 |
| | SON | 0.98 | 0.98 | 0.98 | 2.64 | 19.94 | **2.61** |
| | DJF | 0.99 | 0.99 | 0.99 | 4.01 | **3.76** | 3.80 |
| | Annual | 0.98 | 0.98 | 0.98 | 2.50 | 5.86 | **2.42** |
| | MAM | **0.67** | 0.65 | 0.65 | **34.08** | 34.73 | 34.43 |

| | | | | | | | |
|---|---|---|---|---|---|---|---|
| Sensible Heat Flux | JJA | 0.55 | **0.56** | **0.56** | 30.67 | **30.57** | 30.89 |
| | SON | 0.86 | 0.86 | 0.86 | **23.40** | 25.79 | 23.92 |
| | DJF | **0.88** | 0.87 | 0.87 | **23.71** | 24.42 | 24.42 |
| | Annual | **0.74** | 0.73 | 0.73 | **22.71** | 23.72 | 23.28 |
| Latent Heat Flux | MAM | **0.89** | 0.88 | 0.88 | **15.84** | 16.37 | 16.23 |
| | JJA | **0.82** | **0.82** | 0.81 | 24.24 | **23.18** | 23.40 |
| | SON | 0.88 | 0.88 | 0.88 | 17.34 | 17.57 | **17.33** |
| | DJF | **0.92** | 0.91 | 0.92 | **15.99** | 16.93 | 16.44 |
| | Annual | 0.90 | 0.90 | 0.90 | **13.92** | 14.17 | 14.15 |
| Net Surface Shortwave Flux | MAM | **0.92** | 0.91 | 0.91 | **21.89** | 23.20 | 23.47 |
| | JJA | 0.83 | 0.83 | 0.83 | **29.75** | 29.84 | 30.21 |
| | SON | 0.96 | 0.96 | 0.96 | **20.35** | 26.06 | 21.10 |
| | DJF | 0.96 | 0.96 | **0.97** | **24.28** | 24.51 | 24.32 |
| | Annual | 0.93 | 0.93 | 0.93 | **19.35** | 21.04 | 20.05 |

***Comment 4***:
*Lines 154-155: "It is encouraging that after taking the subgrid energy partitioning into account, the longstanding biases are efficiently mitigated in the EXP_COR run." This is not readily apparent in the bias plots (Fig 2f vs. 2d). Though I see that relative to the CTL (Fig 2e), there is indeed a small decrease in rain rate along the southern edge of the TP, the bias is still over 5 mm/day. It also appears that a dry bias has been re-introduced over Eastern China in EXP_COR that had been mitigated in EXP. A more thorough discussion of where EXP_COR does not improve precipitation is warranted.*

**Reply**: In the CTL run, the wet bias over the southern margin of the TP can exceed 11 mm d$^{-1}$, while that over the eastern margin of the TP is approximately 7 mm d$^{-1}$. Additionally, seen in other CMIP5&6 models, the biases there are much larger than those in the rest of the world (Fig. R8) (Su et al., 2013; Yu et al., 2015; Zhu and Yang, 2020; Lun et al., 2021). In contrast, in the EXP_COR run, the reduced biases over these two regions can be as much as 2.5 mm d$^{-1}$, accounting for a reduction of approximately 25%, especially over the southern margin of the TP (Fig. R9). Given that there are many causes (e.g., unrealistic water vapor advection and the absence of subgrid topographic effects) resulting in the severe overestimation of precipitation along the TP, the improvement in this study, to some extent, is impressive. In the revision, we have adjusted the contour levels and the associated color bar to make the reduced bias in the EXP_COR run noticeable when comparing the biases in the CTL and EXP runs (i.e., using Fig. R8 to replace Fig. 2 and adding Fig. R9 as Fig. 3 in the revision). Additionally, following the suggestion, the degradation in the EXP_COR run compared with the EXP run is discussed in Lines 193-194:

"Over other regions such as southern China, the Middle East and Indonesia, there are some slight degradations in the EXP_COR run compared to the EXP run."

References:

Lun, Y., Liu, L., Cheng, L., Li, X., Li, H., and Xu, Z.: Assessment of GCMs simulation performance for precipitation and temperature from CMIP5 to CMIP6 over the Tibetan Plateau, Int. J. Climatol., 41, 3994-4018, 10.1002/joc.7055, 2021.

Su, F., Duan, X., Chen, D., Hao, Z., and Cuo, L.: Evaluation of the Global Climate Models in the CMIP5 over the Tibetan Plateau, J. Climate, 26, 3187-3208, 10.1175/jcli-d-12-00321.1, 2013.

Yu, R., Li, J., Zhang, Y., and Chen, H.: Improvement of rainfall simulation on the steep edge of the Tibetan Plateau by using a finite-difference transport scheme in CAM5, Clim. Dynam., 45, 2937-2948, 10.1007/s00382-015-2515-3, 2015.

Zhu, Y.-Y. and Yang, S.: Evaluation of CMIP6 for historical temperature and precipitation over the Tibetan Plateau and its comparison with CMIP5, Adv. Clim. Change Res., 11, 239-251, 10.1016/j.accre.2020.08.001, 2020.

[Figure]

**Figure R8.** Spatial distributions of the JJA (June-July August) mean precipitation for (a) TRMM, the biases of (b) CTL, (d) EXP, and (f) EXP_COR with respect to TRMM, and the differences between EXP and CTL (c) and between EXP_COR and CTL (e). The crossed areas are significant at the 95% level. The regionally averaged spatial correlation coefficient (COR) and root mean square error (RMSE) are calculated at the top of (b), (d) and (f).

[Figure]

**Figure R9.** Same as Figure R8 but focusing on the study area (20-50°N, 75-125°E). The regionally averaged spatial COR and RMSE are calculated at the top of (b), (d) and (f).

***Comment 5****:*

*Figure 3: My understanding is that the sum of panels 3a and 3c should match Fig. 2c, and that 3b+3d should equal 2e – is this correct? If so, I wonder if there is a slight plotting mismatch; in Fig 2c, changes in rainfall rarely extend past the TP border, and the small region that does is on the order of 0.5-1 mm/day. But in Fig 3a and 3c, there are precipitation increases that extend along much of the southeastern TP border and into the TP itself, which can be on the order of 1-2 mm/day.*

**Reply**: We apologize for the mismatch. The original Fig. 3 for the EXP run is not correct

because we did not use the same data as in Sun et al. (2021). Figures 3 and 2c&e have been updated by Figs. R10 and R9c&e, respectively (i.e., Figs. 5a-d and 4c&e in the revision).

[Figure]

**Figure R10.** Spatial distribution of the differences in (the first row) large-scale precipitation and (the second row) convective precipitation between (left) EXP and CTL and between (right) EXP_COR and CTL. The crossed areas are significant at the 95% level.

*Comment 6*:
*Lines 200-201: "In contrast, in the EXP_COR run (Fig. 5d), the SLP simulation is corrected…" This does not appear to be the case. The decrease in SLP on the TP in the EXP_COR vs. the CTL case is less than 1 hPa, but the bias in the CTL relative to MERRA2 is more than 10 hPa; the shift is quite small and does not correct the existing bias (though it does reduce it slightly).*

**Reply**: In the revision, "In contrast, in the EXP_COR run (Fig. 5d), the SLP simulation is corrected…" has been deleted, and the corresponding changes are made as follows:

Lines 248-249 in Sect. 3.1:
(1) "Lower (higher) sea level pressure (SLP) anomalies over southern (northern) China are generated in the EXP_COR run than in the EXP run."

Lines 265-266 in Sect. 3.2:
(2) "The above analysis indicates that the precipitation simulation is improved through the adjustment of large-scale atmospheric circulation in the lower atmosphere, which is

highly linked with grid-scale surface heating/cooling (Sun et al., 2021)."

and Lines 301-302:
(3) "As indicated in Sect. 3.1, large-scale atmospheric circulation in the lower atmosphere and local convection are altered as PBL heating changes affect clouds as well."

***Comment 7****:*
*Lines 217-218: "...the underestimations on the southern and eastern margins of the TP in the CTL runs are remarkably improved (Fig. 6f)." When looking for model improvements, it would make more sense to focus on the comparison against observations rather than the control, which would be Fig. 6d. In that light, there is again certainly some improvement in the model bias, but the scale of that change is perhaps minor: on the order of what looks like 2-4 W/m2 when the bias is more than 20 W/m2. The language used in the text suggests a stronger change than is visible.*

**Reply**: We have tuned down the tone in Lines 278-279 in the revision:

"Furthermore, the biases on the southern and eastern margins of the TP and along 60°N in both the CTL and EXP runs are reduced (Fig. 7b-f)."

***Comment 8****:*
*Lines 220-221: "The sensible heat flux changes in the EXP_COR run are more significant than those in the EXP run (Fig. 7f), especially over northern China and the southern and eastern margins of the TP, resulting in better agreement with observations." The change over northern China seems to be stronger in the EXP than the EXP_COR simulation, while the increase in SHFLX over eastern China seems to have actually increased the bias there relative to both the CTL and EXP cases. Focusing in on the southern edge of the TP may miss out on the bigger regional picture in this case.*

**Reply**: Thanks for the comments. Based on a bigger regional picture, both improvements and degradations in the EXP_COR run are discussed in Lines 293-295 in the revised manuscript:

"The positive changes over the southern and eastern margins of the TP in the EXP_COR run are more significant than those in the EXP run (Fig. 8e&f). Nonetheless, we note some degradations from EXP to EXP_COR run (e.g., over southern China)."

***Comment 9:***
*Lines 251-252: "In the EXP_COR run, in addition to the already existing significant improvements in the EXP run…" While it's stated that the EXP run alleviates the overestimation of surface shortwave flux in southern China, it is important to note that the EXP_COR run reduces that improvement and even worsens the bias over eastern*

*China (Fig. 2d). It also seems that the biases in EXP_COR are larger than in EXP over north eastern China/southern Russia, despite general improvements over the TP.*

**Reply**: These degradations in the EXP_COR run have been pointed out in Lines 339-341 in the revision:

"In the EXP_COR run, the underestimation over northern China and the TP in both the CTL and EXP runs is alleviated, although it slightly degrades the simulated shortwave flux over southern China. The negative biases over southeastern Russia in EXP_COR are also larger than those in EXP."

***Comment 10***:
*Figure 12: Unfortunately, this diagram is incredibly hard to read. The legend and key need to use a much larger font size to be readable, and even within the plot itself, distinguishing shapes/numbers from each other is challenging. A table may be easier to read, but at minimum the Taylor diagram needs to be heavily revised. Section 3.3: Differences in the mean state are quite hard to distinguish given the readability issues of Fig. 12.*

**Reply**: Thanks for the suggestion. Table R2-3 has been used instead of Fig. 12 in the revised main text and the supplementary materials (Tables 1 and S1). The related discussion has been revised accordingly in Lines 379-383:

"As presented in Table 1 (global distributions shown in Figs. S3-9), overall, the simulation statistics of the EXP and EXP_COR runs are comparable to those of the CTL run, although slightly different in some seasons. When focusing on East Asia (Table S1), the new schemes outperform the default scheme in terms of COR and RMSE, implying the necessity and importance of parameterizing the subgird land surface heat fluxes to the atmosphere in GCMs in regions with complex terrain (e.g., the TP) and multiple surface types (e.g., eastern China)."

**Table R3.** The COR and RMSE values over the region (20°N-50°N, 75°E-125°E) for the CTL, EXP and EXP_COR runs. MAM is for March-April-May, JJA for June-July-August, SON for September-October-November, and DJF for December-January-February. The best performance among the three experiments is highlighted in bold.

| | | COR | | | RMSE | | |
|---|---|---|---|---|---|---|---|
| | | CTL | EXP | EXP_COR | CTL | EXP | EXP_COR |
| Precipitation | MAM | **0.62** | **0.62** | 0.56 | **2.15** | 2.29 | 2.34 |
| | JJA | 0.48 | **0.60** | **0.60** | 4.51 | 4.07 | **3.71** |
| | SON | **0.56** | 0.51 | 0.55 | **1.89** | 1.92 | 2.01 |
| | DJF | **0.71** | 0.66 | 0.64 | **0.74** | 0.79 | 0.86 |
| | Annual | 0.55 | **0.63** | **0.63** | 2.00 | 1.91 | **1.82** |
| | MAM | 0.95 | 0.95 | 0.95 | **3.09** | 3.14 | **3.09** |
| | JJA | 0.91 | 0.91 | 0.91 | **3.15** | 3.16 | 3.28 |

| | | | | | | | |
|---|---|---|---|---|---|---|---|
| 2 m Temperature | SON | 0.96 | 0.96 | 0.96 | **2.77** | 19.18 | 2.78 |
| | DJF | 0.97 | 0.97 | 0.97 | 4.59 | 4.24 | **4.17** |
| | Annual | 0.96 | 0.96 | 0.96 | 2.89 | 5.93 | **2.87** |
| Sensible Heat Flux | MAM | **0.44** | 0.43 | **0.44** | **34.08** | 34.73 | 34.43 |
| | JJA | 0.55 | **0.60** | 0.53 | 32.96 | **31.54** | 32.63 |
| | SON | -0.01 | 0.02 | **0.06** | 21.68 | 22.21 | **21.33** |
| | DJF | 0.35 | **0.37** | 0.32 | 21.09 | **20.78** | 21.50 |
| | Annual | 0.02 | **0.05** | 0.03 | 25.84 | 25.88 | **25.83** |
| Latent Heat Flux | MAM | 0.72 | 0.73 | **0.76** | 17.69 | 17.27 | **16.92** |
| | JJA | 0.68 | 0.68 | **0.69** | 25.02 | **24.19** | 24.46 |
| | SON | 0.88 | 0.88 | 0.88 | 14.41 | 14.68 | **14.12** |
| | DJF | 0.84 | **0.86** | **0.86** | 9.06 | **8.28** | 8.39 |
| | Annual | 0.83 | 0.83 | **0.84** | 13.34 | 12.93 | **12.80** |
| Net Surface Shortwave Flux | MAM | **0.78** | 0.76 | 0.76 | **25.46** | 27.51 | 25.55 |
| | JJA | 0.69 | **0.76** | 0.71 | 27.28 | **25.95** | 26.38 |
| | SON | **0.66** | 0.63 | 0.61 | **18.78** | 22.21 | 19.78 |
| | DJF | 0.80 | **0.83** | 0.80 | 25.02 | **22.83** | 23.81 |
| | Annual | **0.61** | **0.61** | 0.57 | **18.46** | 18.64 | 18.59 |

***Comment 11***:
*Lines 328-330: "In summary, the performance of the mean state simulations does not change significantly when using the modified scheme (EXP_COR), indicating that the subgrid parameterization scheme can be incorporated into the GCMs without heavy retuning." This may be a bit overstated. It's important to acknowledge that the current analysis is dependent only on annual, global/zonal means, and only a handful of variables. Spatial/seasonal maps could strengthen the arguments made.*

**Reply**: Since Tables R2-3 have summarized the performance of the spatial distributions of seasonal means, we prefer not to show the spatial maps again. As seen in the two tables, for the simulations of precipitation, 2 m air temperature, sensible heat flux, latent heat flux and net surface shortwave flux, the overall performance in EXP_COR does not change much. However, given only a handful of variables as the reviewer indicated, we removed the statement of "indicating that the subgrid parameterization scheme can be incorporated into the GCMs without heavy retuning" in Lines 379-381 in the revision:

"As presented in Table 1 (global distributions shown in Figs. S3-9), overall, the simulation statistics of the EXP and EXP_COR runs are comparable to those of the CTL run, although slightly different in some seasons."

***Comment 12***:
*Lines 382-383: "The mean states did not change much after the introduction of the new parameterization scheme, and thus, the new scheme can be implemented in the CESM without heavy retuning." In addition to the concerns raised above, it's important to note*

*that these experiments are using an outdated version of CESM. In order to be incorporated into the model, these results would need to be reassessed using CESM2; some explanation of why CESM1.2 vs. CESM2 was used could be helpful in this case, and/or whether there are potential challenges to using this within CESM2 vs. CESM1.2.*

**Reply**: Please see our response to the major comment 2 regarding the potential challenges to using this in CESM2. As for why we use CESM1.2 rather than CESM2 in this study, the main reason is that we need to perform the comparison with Sun et al. (2021), which used CESM1.2 as well. We stated this in Line 78 in the revision:

"To be consistent with Sun et al. (2021), the GCM used in this study is the NCAR CESM1.2."

***Minor Comments***
***Comment 1:***
*Lines 34-36: "This results in most of the precipitation simulation errors in GCMs, such as…" Are the authors saying that the use of grid mean rather than subgrid surface fluxes is responsible for the majority of precipitation biases in GCMs? That may be an exaggeration; the studies cited in the rest of the second sentence seem to suggest a range of other ways to mitigate precipitation biases, in fact. Suggest rephrasing for clarity.*

**Reply**: We rephrased the sentence in Lines 37-40 in the revision for clarity:

"This is one of the causes of many precipitation simulation errors in GCMs, such as the bias of the rainfall intensity spectrum (e.g., Dai, 2006; O'Brien et al., 2016; Na et al., 2020; Wang et al., 2021a) and the unrealistic precipitation over the Indian summer monsoon region (e.g., Waliser et al., 2012; Wang et al., 2018) and the eastern and southern parts of the steep Tibetan Plateau (TP) (e.g., Zhou et al., 2021)."

***Comment 2:***
*Line 36: Suggest additional citations on the bias of the rainfall intensity spectrum, as this is a frequently recognized bias that was recognized well before 2021 – the historical context is useful.*

**Reply**: The following references (most of them are published before 2021) have been added to the revised manuscript.

Dai, A.: Precipitation Characteristics in Eighteen Coupled Climate Models, J. Climate, 19, 4605-4630, doi: 10.1175/JCLI3884.1, 2006.

O'Brien, T. A., Collins, W. D., Kashinath, K., Rübel, O., Byna, S., Gu, J., Krishnan, H., and Ullrich, P. A.: Resolution dependence of precipitation statistical fidelity in hindcast simulations, J. Adv. Model Earth Syst., 8, 976-990, 10.1002/2016ms000671, 2016.

Na, Y., Fu, Q., and Kodama, C.: Precipitation Probability and Its Future Changes From a Global Cloud-Resolving Model and CMIP6 Simulations, J. Geophys. Res. - Atmos., 125, 10.1029/2019jd031926, 2020.

***Comment 3:***
*Line 42: "…changes in vegetation density have been found to favor the release of latent heat…" Please be specific in what direction the changes are occurring; does an increase or a decrease in vegetation density link to favored LHFLX?*

**Reply**: It has been specified in Lines 42-45 in the revision:

"For instance, forests dissipate sensible heat to the PBL more efficiently than open landscapes (Rotenberg and Yakir, 2010; Wei et al., 2021), and the increase in vegetation density has been found to favor the release of latent heat rather than sensible heat during the past three and a half decades (Forzieri et al., 2020)."

***Comment 4:***
*Fig 1: Does this spatial pattern of correlation coefficients change with season? It would likely be more useful to the reader to consider DJF/JJA means at least to understand the importance of the correlation coefficient being a time-varying quantity rather than one that is static in time.*

**Reply**: Yes, the spatial pattern of the correlation coefficients varies with time. The DJF/JJA patterns of correlation coefficients (Fig. R11 below) have been added in the revision as Fig. 2. The related changes in the main text are made in Lines 128-136:

"The spatial patterns of the June-July-August (JJA) mean and December-January-February (DJF) mean correlation coefficients are given in Fig. 2b&c. In boreal summer, the sensible and latent heat fluxes in most regions of the world are negatively correlated, except for the TP, Greenland, the central US, and southern Australia (Fig. 2b). In boreal winter, the global distribution is similar to that of the annual mean, showing larger magnitudes of the positive correlation coefficients and smaller magnitudes of the negative correlation coefficients (Fig. 2c). The regions where the correlation coefficients are positive in both summer and winter are mainly located in high latitudes and altitudes. In summer, owing to the melting of snow, latent heat flux increases accordingly as the land surface gains more water for evaporation, and sensible heat flux increases synchronously from enhanced surface net radiation due to increased incoming solar radiation and reduced snow albedo. For winter, decreased solar radiation and increased snow cover reduce both sensible and latent heat fluxes."

[Figure]

**Figure R11.** Spatial distribution of (a) annual, (b) JJA (June-JulyAugust) and (c) DJF (December-January-February) mean correlation coefficients *r* between the subgrid surface sensible heat and latent heat fluxes in the EXP_COR simulation.

**Comment 5:**
*Line 123: Please clarify the time scale at which the correlation coefficient r varies; is this computed at each time step?*

**Reply**: Yes, it varies at each time step. We clarified this in Lines 118-124 in the revision:

"However, we can compute the correlation coefficients between the subgrid sensible and latent heat fluxes within each grid cell at every time step (i.e., 30 minutes) using the following equation:

$$r = \frac{\sum_{i=1}^{n} w_i (F_{SH_i} - \bar{F}_{SH})(F_{LH_i} - \bar{F}_{LH})}{\sigma_{SH} \sigma_{LH}} \quad (2)$$

where $n$ is the number of PFTs within a grid cell in the land model; $w_i$ is the area percentage of each PFT within the grid cell; $F_{SH_i}$ and $F_{LH_i}$ are the subgrid surface sensible and latent heat fluxes of each PFT, respectively; $\bar{F}_{SH}$ and $\bar{F}_{LH}$ are the weighted averages of the subgrid sensible and latent heat fluxes in one grid cell, respectively; and $\sigma_{SH}$ and $\sigma_{LH}$ are the standard deviations of the subgrid sensible and latent heat fluxes in one grid cell, respectively."

**Comment 6:**

*Section 2.3: Are the seasonal averages based on monthly means or higher temporal resolution? Daily data may yield a more realistic picture of model performance, even though it is averaged up to the seasonal level in analysis.*

**Reply**: All the analyses are computed based on the monthly output. For a single variable ($x$) (e.g., temperature and precipitation), the seasonal averages using the monthly output are consistent with those using the daily output (both averaged over the model time step, 30 min). This is because the daily perturbation ($x'$) on the multiyear seasonal averages should be 0 (i.e., $\bar{x'} = 0$).

**Comment 7:**

*Figure 2: Since the main region of interest in the TP, it would be helpful to also give the RMSE and correlation for that specific region (as it seems may be the norm in other figures of this paper).*

**Reply**: Thanks for the valuable suggestion. The RMSE and COR for the TP region have been added to Fig. 4 in the revision.

**Comment 8:**

*Figure 3 (and other figures): Suggest adding labels that denote "EXP – CTL" and "EXP_COR – CTL" to each subpanel, as in Fig 2.*

**Reply**: Done.

**Comment 9:**

*Lines 196-200: This section seeks to draw comparisons to the EXP and EXP_COR simulations, but there are no EXP-related plots in Fig. 5; please reproduce figures from Sun et al. that are key to this analysis within the current paper.*

**Reply**: Done. Please see Fig. R12 below (Fig. 6 in the revision).

[Figure]

**Figure R12**. Spatial distributions of the differences of JJA-mean PBL heating (a) between EXP and CTL, and (b) between EXP_COR and CTL, JJA-mean SLP superposed by the vector $\vec{V}$ from (c) MERRA-2, and the differences (d) between CTL and MERRA-2, (e) between EXP and CTL, (f) between EXP and MERRA2, (g) between EXP_COR and CTL, and (h) between EXP_COR and MERRA2. The vector $\vec{V}$ is defined in Eq. (3).

[Figure]

**Figure R13.** Spatial distributions of the JJA-mean SLP superposed by the vector $\vec{V}$ from (a) MERRA2, (b) CTL, (c) EXP and (d) EXP_COR. The vector $\vec{V}$ is defined in Eq. (3).

*Comment 10:*
*Figure 5: In addition to including plots of the EXP case, it would be helpful to also include the raw EXP_COR analysis and (space permitting for this) the difference from MERRA2 observations and not just the CLT case. This would make the analysis points that are made much more convincing and clearer.*

**Reply**: The raw EXP_COR analysis has been added to Fig. R13 (Fig. S2 in the supplementary material). The differences in EXP and EXP_COR between MERRA2 observations are included in Fig. R12d, f and h (Fig. 6d, f, h in the revision).

*Comment 11:*
*Line 211: As noted above, it does not appear that the atmospheric circulation is "corrected" in this case, so I suggest softening the language.*

**Reply**: We tuned down the tone in Lines 25-27 in the revision:

(1) "The altered large-scale circulation in the lower atmosphere due to anomalous heating/cooling in the planetary boundary layer is responsible for the change in moisture transport."

and Lines 265-266:

(2) "The above analysis indicates that the precipitation simulation is improved through the adjustment of large-scale atmospheric circulation in the lower atmosphere, which is highly linked with grid-scale surface heating/cooling (Sun et al., 2021)."

and Lines 301-302:

(3) "As indicated in Sect. 3.1, large-scale atmospheric circulation in the lower atmosphere and local convection are altered as PBL heating changes affect clouds as well."

***Comment 12:***
*Figure 6 (and others): Please clarify if the RMSE and COR values are just for the inset region over the TP? It would be helpful in general to have these values for both the full domain and the TP region itself.*
*It would also result in a cleaner plot to simply place a rectangle around the TP region where those values are calculated (perhaps in panel a) rather than an additional inset plot since the TP region is not any larger than in the main subpanel.*

**Reply**: We have updated Figs. 6-11 (now Figs. 7-12 in the revision) with a global view. Following the suggestion of reviewer #2, the inset plots of the TP region in each figure have been removed. The RMSE and COR are calculated for both the full domain and the TP region itself. The values for the latter are summarized separately in Table R3, which has been included in the supplementary material as Table S1.

***Comment 13:***
*Line 269: "It should be noted that there is a significant negative band along 60°N."*
*More explanation should be included if this is to be noted – why is this band present in EXP_COR but not EXP, does this suggest better/worse agreement with obs, etc.*

**Reply**: This is discussed in detail in Lines 292-293 in the revision:

"On top of the EXP run, EXP_COR further alleviates the overestimation along 45°N-60°N over the Eurasian continent where sensible heat fluxes and latent heat fluxes are highly correlated in this region (Fig. 2b)."

***Comment 14:***
*Fig 10: In this figure and similar ones that follow, it would be useful to show the raw CTL case values as well so that the reader can determine how large each change is relative to the baseline. It would also again be useful to include "EXP-CTL" and "EXP_COR-CTL" in the plot titles.*

**Reply**: The simulations in the CTL run have been added to those figures (now Figs. 9-10 in the revision). The titles "EXP-CTL" and "EXP_COR-CTL" were also included.

***Comment 15:***
*Fig. 13: It would be helpful to show the difference in bias (relative to obs) for each case, not just the difference in each case relative to CTL. That would make the statements in section 3.3 more clearly supported.*

**Reply**: The differences in biases relative to the observations for the three simulations

are shown in Figure R14 (now Fig. 13 in the revision). The difference in each case relative to CTL was retained in Fig. S8 (Fig. R15 here) in the revision for reference.

[Figure]

**Figure R14.** Annual and zonal mean cross-sections of the (a–c) temperature and (d–f) specific humidity differences for (a&d) CTL-ERAI, (b&e) EXP-CTL, and (c&f) EXP_COR-CTL. The crossed areas are significant at the 95% level.

[Figure]

**Figure R15.** Annual and zonal mean cross-sections of the (a–c) temperature and (d–f) specific humidity differences for (a&d) CTL-ERAI, (b&e) EXP-CTL, and (c&f) EXP_COR-CTL. The crossed areas are significant at the 95% level.

*Comment 16:*
*Lines 369-372: The precipitation improvements from EXP are mostly present in EXP_COR, but Fig. 2f suggests a reintroduction of a dry bias over eastern China that had been removed in EXP, which should be noted. The overestimates in precipitation along the southern/eastern borders of the TP are also still present and of a large amplitude, so with the current colorbar it is hard to deduce that those biases are "significantly alleviated."*

**Reply**: Please see our response to major comment 4.

---

## Author Comment (AC2)

**Reply to the comments by Referee #2**

We thank the reviewer for the comments and suggestions to improve our manuscript. Below is our point-by-point response to these comments. The reviewer's comments are in italics, our responses are in normal font, and manuscript revisions are in blue.

***General comments***
***Comment 1****:*
*The authors briefly summarized the subgrid surface heat flux scheme initially proposed by Sun et al. (2021) and specifically introduced the modified version. These two stochastic sampling models actually introduce the uncertainties into the land-atmosphere coupling process other than, as the authors claimed, represent the realistic surface heat flux partitioning. How does the model deal with the surface energy balance closure in accompany with the imposed heat flux partitioning? On the assumption of normal distribution of the surface heat fluxes, how many land grid cells are subject to that normal distribution if the purpose of this study is to parameterize the realistic surface heat fluxes? And to what extent the other "abnormal" grid cells can play roles in impacting the simulated climate? I understand that it may be difficult to address these points at one time, but it will benefit the readers if the authors can provide some advantages and disadvantages of Sun's approach and the modified approach in the text.*

**Reply**: Thanks for the insightful comments. The surface energy balance closure **at the grid scale** is not affected by both the stochastic sampling method and the follow-up collocation of the sampled sensible and latent heat fluxes according to their correlation coefficient **at the subgrid scale**. **Note that the surface energy balance has been closed at the grid scale in the default land-atmosphere coupling way.** Therefore, the stochastic sampling at the subgrid scale based on the truncated normal distributions with the mean values equal to the default grid averages calculated by the weighted fluxes on each PFT within the grid cell can assure the grid-scale surface energy balance closed as well in the long-term statistics, although at a given time step this might be broken up. This is confirmed by Fig. R1 where the distributions of sampled sensible heat fluxes highly resemble the realistic distributions of sensible heat fluxes within the given grid cells in the long-term statistics. As a result, their grid averages are comparable. This feature is not affected by the follow-up collocation of the sampled sensible and latent heat fluxes because this process does not alter the sampled values just arranging them in a given sequence. We clarified this in Lines 110-114 in the revision:

"Note that the surface energy balance closure at the grid scale is not affected by the stochastic sampling method. The surface energy balance has been closed at the grid scale in the default land-atmosphere coupling way. Therefore, the stochastic sampling at the subgrid scale based on the truncated normal distributions with the mean values equal to the default grid averages calculated by the weighted fluxes on each PFT within

the grid cell (Fig. 1) can assure the grid-scale surface energy balance closed as well in the long-term statistics, although at a given time step this might be broken up."

[Figure]

**Figure R1.** The histogram and gaussian kernel density estimate (KDE) (dashed line) of the sensible heat fluxes at the PFT in the eight grid cells with 16 (top row) and 8 (bottom row) PFTs, respectively. Green histograms and KDE estimates show the distribution of realistic sensible heat fluxes at the PFT within each grid cell while purple histograms and KDE estimates are for the distribution of sampled sensible heat fluxes based on the assumed normal distribution.

[Figure]

**Figure R2.** Number of PFTs in each grid cell. "A*" and "B*" denote the grid cell with 16 and 8 PFTs, respectively.

To verify the validity of the assumed normal distribution, we randomly selected land grids where the subgrid PFTs are as many as possible because, with fewer subgrid PFTs within the grid cell, the diversity of the realistic fluxes (associated with extreme conditions) reduces. Figure R1 shows the results over those grid cells with the maximum number of PFTs (i.e., 16) and the half (i.e., 8) both covering climate regimes as many as possible (Fig. R2). We can see that no matter for 16 PFTs or 8, the distributions of sampled sensible heat fluxes highly resemble the realistic distributions within the given grid cells in the long-term statistics. Therefore, the assumed normal distribution works well and thus the sampled samples can represent the realistic features for climate simulation. Figure R1 has been included in the revised manuscript as Fig. 1 and Fig. R2 in the supplementary materials as Fig. S1. This point was clarified as well in Lines 96-104 in the revision:

"The stochastic sampling implicitly parameterized the uncertainties of the PBL and convection processes to a certain degree. As stated in Sun et al. (2021), using the sampled fluxes from a statistical distribution rather than the fluxes directly from individual PFTs can represent the mix of subgrid fluxes from horizontally mixed land cover types in reality. Moreover, the distribution of the sampled subgrid surface heat fluxes based on the assumed normal distribution resembles the distribution of realistic subgrid PFT heat fluxes within the grid cell in long-term statistics. As shown in Fig. 1 for the sensible heat flux, over the grid cells with 16 and 8 PFTs, the two distributions are highly consistent, in terms of mean value, variance and skewness. The latent heat flux has similar results (figure not shown). Given that those grid cells are stochastically selected and cover different climate regimes (Fig. S1), therefore, the assumed normal distribution works well and thus the sampled samples can represent the realistic features for climate simulation."

*Comment 2*:
*As the manuscript reads now, it seems like the authors touched on a wide range of analysis of the atmospheric variables only briefly without really figuring out their scientific connections. Instead, the authors may consider restructuring the manuscript and dig deeper into aspects that are truly linked with the incorporation of the subgrid-scale treatment of surface heat fluxes. For example, the authors stated that the simulated large-scale circulation is significantly altered by the modified surface heat fluxes which further improves the simulated precipitation on the southern and eastern margins of TP. If so, is it possible to demonstrate the relationship between the changes in surface heat fluxes (Figs. 6&7) and the affected SLP field in Fig. 5? Also, it is very interesting to see the cloud feedbacks in EXP or EXP_COR, and that is supposed to be induced by the surface energy changes in this study. But in the context, the changes in cloud properties are considered the reason why surface heat fluxes are influenced.*

**Reply**: Thanks for the valuable comments. The scientific connections among those variables (e.g., precipitation, surface energy fluxes, clouds and 2 m temperature) have been strengthened in the revision with a schematic diagram (Fig. R3) of figuring out

their physical links included. Also, the manuscript has been restructured accordingly, especially in Sect. 3.

Over eastern China and the eastern/southern borders of the TP, the changes in the PBL heating rate from EXP to EXP_COR (Fig. R4a&b) are analyzed to link the changes in surface heat fluxes with the affected SLP. The scientific connections between variables are disentangled as follows (Fig. R3a). As stated in Sun et al. (2021), in the EXP run, the subgrid variations of the land surface heat fluxes increase (decrease) the PBL heating rate over southern (northern) China. After taking the partitioning of subgrid surface heat fluxes into account, the increase (decrease) in the PBL heating rate over southern (northern) China is strengthened (Fig. R4b). Therefore, the destabilization (stabilization) in the lower atmosphere is further enhanced, promoting (suppressing) the development of local convection. Lower (higher) sea level pressure (SLP) anomalies over southern (northern) China are generated in the EXP_COR run than in the EXP run (Fig. R4g). In particular, compared with the EXP run, the anomalous high SLP over northern China slightly extends to the south and the anomalous low SLP over southern China retreats. The anomalous anticyclone over northern China expands accordingly, which engenders decreased precipitation on the eastern border of the TP and a slight dry bias over southern China. Convective precipitation dominates the changes of total precipitation over eastern China and the eastern margin of the TP. In the EXP_COR run, the easterly anomaly along 25° N-30° N partly blocks moisture transport from the ocean in the south to the southern margin of the TP, and therefore, the decrease of large-scale precipitation dominates the change of precipitation simulation on the southern margin of the TP.

As the reviewer indicated, we corrected the statement that the changes in clouds should be a result of the altered PBL heating rates and the associated local convection, although they in turn can affect the net surface heat fluxes. This is summarized in Fig. R3b.

In the revision, Figs. R3&4 are included as Figs. 14&6, respectively and the related edits are made in Lines 455-462:

"The causes are briefly summarized in Fig. 14a. The subgrid variations of the land surface heat fluxes increase (decrease) the PBL heating over southern (northern) China. With the further introduction of the partitioning of subgrid surface heat fluxes, the increase (decrease) in the PBL heating over southern (northern) China is elevated, thus destabilizing (stabilizing) the lower atmosphere. Resultantly, local convection is promoted (suppressed) over southern (northern) China. The changes of convective precipitation dominate the changes of total precipitation over eastern China and the eastern margin of the TP. The altered large-scale circulation associated with the easterly anomaly along 25° N-30° N partly blocks moisture transport from the ocean in the south to the southern margin of the TP. Accordingly, the decrease of large-scale precipitation is responsible for the reduced precipitation there."

and in Lines 463-467:

"The links among clouds, net surface shortwave flux and 2 m air temperature over eastern China are figured out in Fig. 14b. As the PBL heating decreases in northern China, the lower atmosphere stabilizes and local convection is suppressed. Accordingly, middle and high clouds, and associated CWP decrease (Figs. 9&10). Thus, SWCF decreases over northern China, which increases the net surface shortwave flux. As the surface gains more energy, the near-surface air temperature warms. In contrast, southern China features the opposite changes in the storyline."

[Figure]

**Figure R3.** Schematic diagram of summarizing the climate impacts of parameterizing subgrid variations and partitioning of land surface heat fluxes to the atmosphere.

[Figure]

**Figure R4.** Spatial distributions of the differences of JJA-mean PBL heating (a) between EXP and CTL, and (b) between EXP_COR and CTL, JJA-mean SLP superposed by the vector $\vec{V}$ from (c) MERRA-2, and the differences (d) between CTL and MERRA-2, (e) between EXP and CTL, (f) between EXP and MERRA2, (g) between EXP_COR and CTL, and (h) between EXP_COR and MERRA2. The vector $\vec{V}$ is defined in Eq. (3).

***Specific comments***
***Comment 1:***
*L75: Suggest changing the title to "Materials and methods" or "Methodology"*
**Reply**: Done.

**Comment 2:**

*L82: "all of the PFTs in the grid" -> "the corresponding grid"*
**Reply**: Done.

**Comment 3:**
*L88: "selected as N pairs" -> "paired with each other"*
**Reply**: Done.

**Comment 4:**
*L97: "normal distributions" see the general comment*
**Reply**: Please see our response to the general comment 1.

*Comment 5:*
*L152: "Sun et al. improved ... (Fig. 2c)" Should it be Fig. 2d or Fig. 2c? Fig. 2c is comparing Sun et al. results with the control case.*

**Reply**: Sorry for this typo. It should be Fig. 2d (now Fig. 3d in the revision).

*Comment 6:*
*L157: Given that the RMSE of EXP_COR is very close to EXP, is it sensitive to the average region the authors defined? Or, why do the authors choose 0-50°N, 0-180°E as the study area for the statistics?*

**Reply**: Following reviewer #1's suggestion, Fig. 2 (now Fig.3) has been expanded to -48.5°S-48.5°N, -180°W-180°E to agree with the coverage area of the TRMM observations.

*Comment 7:*
*L198-L200: If the authors can include the Fig.5b from Sun et al. (2021) into Fig. 5 in this manuscript, it may be easier for readers to understand the context on explaining model's overestimated precipitation in the southern TP.*

**Reply**: Fig. 5a&b in Sun et al. (2021) has been included in the figure (now Fig. 6a&e in the revision).

*Comment 8:*
*Figure 5: Does it show the moisture transport vector or wind speed? The unit as depicted in Fig. 5 is m s⁻¹.*

**Reply**: This is the moisture transport vector ($\vec{V}$) as shown in Eq. (2) (now Eq. (3) in the revision), which is given by $W^{-1} \int_{P_{top}}^{P_{bot}} (q\vec{u}) dp/g$, representing the total horizontal moisture transport normalized to the column integrated moisture. Therefore, the unit is m s⁻¹. For clarity, we explained this more in Lines 227-229 in the revision:

"The vector $\vec{V}$ with the units of m s$^{-1}$, given by $W^{-1}\int_{P_{top}}^{P_{bot}}(q\vec{u})dp/g$, represents the total horizontal moisture transport normalized to the column-integrated moisture, where $\vec{u}$ is the horizontal wind vector."

and in Line 263:

"The vector $\vec{V}$ is defined in Eq. (3)."

***Comment 9:***
*Figures 6-11: The inserted boxes do not zoom in much or indicate more useful information. They may be removed.*
**Reply**: Done.

***Comment 10:***
*L221: Remove "resulting in better agreement with the observations"*
**Reply**: Done.

***Comment 11:***
*L303-L305: This sentence is a little confusing. What is the difference between "subgrid variations in surface heat fluxes" and "their subgrid partitioning"? It might be good to provide an explanation.*

**Reply**: The sentence has been rephrased in Lines 374-377 in the revision:

"The precipitation improvements over eastern China are mainly from the consideration of subgrid variations in surface heat fluxes (i.e., the EXP run where the sampled subgrid sensible and latent heat fluxes are stochastically paired with each other), while the improved precipitation simulations on the southern and eastern margins of the TP are attributed to the further inclusion of the partitioning of the subgrid surface heat fluxes (the EXP_COR run)."

***Comment 12:***
*Figure 12: Could the authors make this blurry plotting of a higher quality?*

**Reply**: Following review #1's suggestion, Fig.12 has been translated into Table 1 (Table R1 below) for clarity additionally with the seasonal statistics included.

**Table R1.** The COR and RMSE values in the CTL, EXP and EXP_COR runs. MAM is for March-April-May, JJA for June-July-August, SON for September-October-November, and DJF for December-January-February. The best performance among the three experiments is highlighted in bold.

| Variables | Period | COR | RMSE |
| --- | --- | --- | --- |

| | | CTL | EXP | EXP_COR | CTL | EXP | EXP_COR |
|---|---|---|---|---|---|---|---|
| Precipitation | MAM | **0.82** | **0.82** | 0.81 | **1.55** | **1.55** | 1.61 |
| | JJA | 0.78 | **0.80** | 0.79 | 2.11 | **2.03** | 2.04 |
| | SON | 0.85 | 0.85 | 0.85 | 1.53 | **1.52** | 1.53 |
| | DJF | **0.85** | 0.84 | 0.84 | **1.62** | 1.65 | 1.73 |
| | Annual | 0.86 | 0.86 | 0.86 | 1.29 | **1.27** | 1.30 |
| 2 m Temperature | MAM | 0.98 | 0.98 | 0.98 | 2.57 | 2.50 | **2.49** |
| | JJA | 0.95 | 0.95 | 0.95 | 2.70 | **2.66** | 2.67 |
| | SON | 0.98 | 0.98 | 0.98 | 2.64 | 19.94 | **2.61** |
| | DJF | 0.99 | 0.99 | 0.99 | 4.01 | **3.76** | 3.80 |
| | Annual | 0.98 | 0.98 | 0.98 | 2.50 | 5.86 | **2.42** |
| Sensible Heat Flux | MAM | **0.67** | 0.65 | 0.65 | **34.08** | 34.73 | 34.43 |
| | JJA | 0.55 | **0.56** | **0.56** | 30.67 | **30.57** | 30.89 |
| | SON | 0.86 | 0.86 | 0.86 | **23.40** | 25.79 | 23.92 |
| | DJF | **0.88** | 0.87 | 0.87 | **23.71** | 24.42 | 24.42 |
| | Annual | **0.74** | 0.73 | 0.73 | **22.71** | 23.72 | 23.28 |
| Latent Heat Flux | MAM | **0.89** | 0.88 | 0.88 | **15.84** | 16.37 | 16.23 |
| | JJA | **0.82** | **0.82** | 0.81 | 24.24 | **23.18** | 23.40 |
| | SON | 0.88 | 0.88 | 0.88 | 17.34 | 17.57 | **17.33** |
| | DJF | **0.92** | 0.91 | 0.92 | **15.99** | 16.93 | 16.44 |
| | Annual | 0.90 | 0.90 | 0.90 | **13.92** | 14.17 | 14.15 |
| Net Surface Shortwave Flux | MAM | **0.92** | 0.91 | 0.91 | **21.89** | 23.20 | 23.47 |
| | JJA | 0.83 | 0.83 | 0.83 | **29.75** | 29.84 | 30.21 |
| | SON | 0.96 | 0.96 | 0.96 | **20.35** | 26.06 | 21.10 |
| | DJF | 0.96 | 0.96 | **0.97** | **24.28** | 24.51 | 24.32 |
| | Annual | 0.93 | 0.93 | 0.93 | **19.35** | 21.04 | 20.05 |

---

## Referee Report (RR1)

**Reviewer's report for gmd-2022-114-R1**

The authors have largely addressed my comments in the original review, and their editorial modifications have clarified most of the paper's results. But meanwhile, the authors included some discussions absent in the original manuscript, which raises additional questions as below.

**Comments**

L140-142: "improvements are made to Sun scheme" It might be inappropriate to directly say the modified scheme is an improvement. -> "We propose two methods below based on the subgrid surface energy partitioning between sensible and latent heat fluxes"

L152: "input" -> "passed"

L155: "this process does not alter the sampled subgrid values just arranging them in a given sequence" -> "this process just rearranges the sequence of heat fluxes rather than altering their values"

L159: "One" -> "The"

L160: "another" -> "the"

L161: "the third ..." -> "the EXP_COR uses the modifications as described in Sect. 2.2"

L179-180: "one improvement we expected in the new scheme is the alleviation of the overestimated summer precipitation on the southern and eastern margins of the Tibetan Plateau" Why should we expect the precipitation bias will be reduced on the southern and eastern margins of TP? It is clueless for readers to get this point at this place. Rephrase this paragraph for clarity.

L185-186: "The simulated precipitation over Arabia and Indonesia is improved as well while that over the southeastern US is degraded." This sentence may be removed cause there are barely any differences between Fig. 3c and 3e, as well as Fig. 3b, 3d, and 3f, except that the improved precipitation in eastern China is visible in EXP.

L190: "the improved boreal summer precipitation over eastern China" It seems like the EXP_COR does not simulate better precipitation than EXP over eastern China as we see negative biases over there in Fig. 3f.

L195: Fig. 3 - It looks like the changes induced by EXP and EXP_COR are minor, particularly at a global scale. Does a figure of relative bias difference (e.g., (EXP-CTL)/(CTL-TRMM)) help if the authors want to highlight the improvement on the southern and eastern margins of TP? Also, the Figs. 7-12 in this manuscript have similar issues as in Fig. 3. See the comments as follows.

L206-208: The values of RMSE and the correlation coefficient in the context differ from Fig. 4.

L213-214: "both large-scale precipitation and convective precipitation slightly increase on the southern border of the TP" If so, an increase in the total precipitation should be seen in the southern margin of TP from Fig. 3c. It's apparently not the case here.

L214-215: "large-scale precipitation increases" It appears that the large-scale precipitation increases in the southeastern but decreases in the northeastern. In fact, the EXP run mostly improves the simulated precipitation over eastern China and has little impact on TP margins. Maybe removing the context regarding EXP TP margins is more acceptable than an inaccurate description.

L250-251: Why does the anticyclone over northern China decrease the precipitation on the eastern border of TP?

L264: Section 3.2 could be shortened to be more concise and organized. As it reads now, this section does not make much sense for improving the mechanistic understanding of climate response to the proposed surface heat flux parameterization. Most of the comparisons between CTL versus EXP/EXP_CORE or observations versus simulations are not distinctly shown in the corresponding plots, such that the analysis in this part is a bit vague and unconvincing. For example, in L275, the authors stated that the positive biases over southern China are reduced, but Fig. 7c and 7d are almost identical to each other and that improvement is not noticeable actually. In L339, "the underestimation over northern China and the TP in both the CTL and EXP runs is alleviated in EXP_COR", but Fig. 11d even shows darker green than Fig. 11c for northern China (i.e., EXP_COR has more negative bias than EXP in net surface shortwave flux).

L307: "larger" -> "large"

L319: "distributions of the CTL run" -> "distributions of the low, middle, and high clouds in the CTL run"

L371: "improves the simulations of summer precipitation in Asia" Does EXP or EXP_COR improve the precipitation in Asia? It seems like only for eastern China in EXP and TP area in EXP_COR.

L398-399: What values? Fig. S8 depicts the vertical structure of clouds.

L433-435: "Compared with MAM4 ... cloud macrophysics schemes in CAM6" Needs to be clarified. Are the authors comparing EXP_COR with physics parameterizations or comparing the physics parameterizations from different CESM versions?

Have a native speaker assist with the writing.

---

## Author Response (AR2)

**Reply to the comments by Referee #2**

We thank the reviewer for the comments and suggestions to further improve our revised manuscript. Below is our point-by-point response to these comments. The reviewer's comments are in italics, our responses are in normal font, and manuscript revisions are in blue.

*The authors have largely addressed my comments in the original review, and their editorial modifications have clarified most of the paper's results. But meanwhile, the authors included some discussions absent in the original manuscript, which raises additional questions as below.*

**Reply**: We thank the reviewer for the positive remarks on our efforts to improve the original manuscript. Please see our response to the additional questions below.

*Comments*
*Comment 1:*
*L140-142: "improvements are made to Sun scheme" It might be inappropriate to directly say the modified scheme is an improvement. -> "We propose two methods below based on the subgrid surface energy partitioning between sensible and latent heat fluxes"*

**Reply**: Done.

**Comment 2:**
*L152: "input" -> "passed"*

**Reply**: Done.

**Comment 3:**
*L155: "this process does not alter the sampled subgrid values just arranging them in a given sequence" -> "this process just rearranges the sequence of heat fluxes rather than altering their values"*

**Reply**: Done.

**Comment 4:**
*L159: "One" -> "The"*

**Reply**: Done.

*Comment 5:*
*L160: "another" -> "the".*

**Reply**: Done.

*Comment 6:*
*L161: "the third ..." -> "the EXP_COR uses the modifications as described in Sect. 2.2"*

**Reply**: Done.

*Comment 7:*
*L179-180: "one improvement we expected in the new scheme is the alleviation of the overestimated summer precipitation on the southern and eastern margins of the Tibetan Plateau" Why should we expect the precipitation bias will be reduced on the southern and eastern margins of TP? It is clueless for readers to get this point at this place. Rephrase this paragraph for clarity.*

**Reply**: We rephrased the paragraph in Lines 165-168 in the revision for clarity:

"Sun et al. (2021) found that the improved precipitation simulation with the parameterization of subgrid surface heat fluxes to the atmosphere is most prominent for boreal summer. Therefore, to compare with Sun et al. (2021), the analyses are first focused on boreal summer followed by a thorough evaluation of the two parameterizations on simulated climate variables for four seasons at the global scale."

*Comment 8:*
*L185-186: "The simulated precipitation over Arabia and Indonesia is improved as well while that over the southeastern US is degraded." This sentence may be removed cause there are barely any differences between Fig. 3c and 3e, as well as Fig. 3b, 3d, and 3f, except that the improved precipitation in eastern China is visible in EXP.*

**Reply**: Done.

*Comment 9:*
*L190: "the improved boreal summer precipitation over eastern China" It seems like the EXP_COR does not simulate better precipitation than EXP over eastern China as we see negative biases over there in Fig. 3f.*

**Reply**: The indicated improvements in the sentence are relative to the CTL simulation rather than the EXP run. To avoid confusion, we removed this sentence in the revision.

*Comment 10:*
*L195: Fig. 3 - It looks like the changes induced by EXP and EXP_COR are minor, particularly at a global scale. Does a figure of relative bias difference (e.g., (EXP-CTL)/(CTL-TRMM)) help if the authors want to highlight the improvement on the southern and eastern margins of TP? Also, the Figs. 7-12 in this manuscript have*

*similar issues as in Fig. 3. See the comments as follows.*

**Reply**: Following the suggestion, the global distributions of the differences in the relative biases of EXP and EXP_COR are shown in Fig. R1. There are many regions (especially over oceans) showing dramatic relative changes. However, most of them are uninformative and unrealistic, which are magnified by the extremely small biases in CTL (CTL - TRMM) although the absolute changes in EXP and EXP_COR (EXP – CTL and EXP_COR - CTL) are quite minor (Fig. 3 in the main text). Thus, a demonstration of the relative bias difference largely screens out informative signals in the regions with absolute biases largely reduced.

To make the reviewer convinced that the reduction of the relative biases on the southern and eastern margins of the TP can be approximately 25% (a considerable improvement for the absolute biases up to 11 mm d$^{-1}$ there in CTL), which is indicated in the revised manuscript (Lines 186-188), Fig. R2, same as Fig. R1, but zooms in on the region (20-50°N, 75-125°E). It is clear that the biases over the southern and eastern margins of the TP decreased by approximately 25% - 50%. Given that this relative improvement is feasibly derived from the absolute biases in Fig. 4c&e in the main text, we prefer not to add Figure R2 in the revision.

Figures 7-12 in Sect. 3.2 aim to present a thorough evaluation of the performance of two parametrizations on simulated climate variables rather than particularly seeking significant improvements to all the variables in the EXP_COR run compared to the EXP run. Therefore, the comparable performance between EXP and EXP_COR is acceptable. We have revised the discussion regarding Figs. 7-12 in the revision. See our response to the comments below in detail.

[Figure]

**Figure R1.** Global distributions of the differences in the biases of EXP and EXP_COR relative to CTL.

[Figure]

**Figure R2.** Same as Fig. R1, but focusing on the study area (20-50°N, 75-125°E).

*Comment 11:*
*L206-208: The values of RMSE and the correlation coefficient in the context differ from Fig. 4.*

**Reply**: Thank you very much for pointing out this mismatch. The values of RMSE and correlation coefficient in the context are calculated over the region (20°N-50°N, 75°E-

125°E), while those in Fig. 4 are for a slightly different region (i.e., 20°N-48°N, 75°E-125°E). In the revision, their calculations have been unified to the same region (20°N-50°N, 75°E-125°E), and Fig. 4 has been updated correspondingly (Fig. R3 below).

[Figure]

**Figure R3.** Spatial distributions of the JJA (June-July-August) mean precipitation in the region (20-50°N, 75-125°E) for (a) TRMM, the biases of (b) CTL, (d) EXP, and (f) EXP_COR with respect to TRMM, and the differences between EXP and CTL (c) and between EXP_COR and CTL (e). The crossed areas are significant at the 95% level. The regionally averaged spatial correlation coefficient (COR) and root mean square error (RMSE) are given at the top of (b), (d) and (f).

***Comment 12:***
*L213-214: "both large-scale precipitation and convective precipitation slightly increase on the southern border of the TP" If so, an increase in the total precipitation*

*should be seen in the southern margin of TP from Fig. 3c. It's apparently not the case here.*

**Reply**: Following Comment 13, the context regarding EXP TP margins is removed.

***Comment 13:***
*L214-215: "large-scale precipitation increases" It appears that the large-scale precipitation increases in the southeastern but decreases in the northeastern. In fact, the EXP run mostly improves the simulated precipitation over eastern China and has little impact on TP margins. Maybe removing the context regarding EXP TP margins is more acceptable than an inaccurate description.*

**Reply:** Done.

***Comment 14:***
*L250-251: Why does the anticyclone over northern China decrease the precipitation on the eastern border of TP?*

**Reply**: Higher sea level pressure anomalies over northern China are generated in EXP_COR than in EXP (Fig. 6e&g in the main text). In particular, compared with the EXP, the anomalous high SLP over northern China extends further to the south and the eastern border of the TP. The resulting downdraft associated with the anomalous anticyclone causes decreased precipitation there. The decrease of precipitation on the eastern border of the TP is directly from the further westward expansion of the anomalous anticyclone over northern China rather than indirectly from a remote effect of the anomalous anticyclone over northern China. In this revision, we clarified the discussion in Lines 230-234:

"In particular, compared with the EXP run, the anomalous high SLP over northern China extends further to the south as well as the eastern border of the TP with the anomalous low SLP over southern China retreating (Fig. 6d-h). The anomalous anticyclonic moisture transport associated with downdraft expands accordingly, which engenders decreased precipitation on the eastern border of the TP and slight dry biases over southern China."

***Comment 15:***
*L264: Section 3.2 could be shortened to be more concise and organized. As it reads now, this section does not make much sense for improving the mechanistic understanding of climate response to the proposed surface heat flux parameterization. Most of the comparisons between CTL versus EXP/EXP_COR or observations versus simulations are not distinctly shown in the corresponding plots, such that the analysis in this part is a bit vague and unconvincing.*
*For example, in L275, the authors stated that the positive biases over southern China are reduced, but Fig. 7c and 7d are almost identical to each other and that improvement*

*is not noticeable actually. In L339, "the underestimation over northern China and the TP in both the CTL and EXP runs is alleviated in EXP_COR", but Fig. 11d even shows darker green than Fig. 11c for northern China (i.e., EXP_COR has more negative bias than EXP in net surface shortwave flux).*

**Reply**: Thank you for the suggestion. Section 3.2 has been shortened in the revision.

***Comment 16:***
*L307: "larger" -> "large"*

**Reply**: Done.

***Comment 17:***
*L319: "distributions of the CTL run" -> "distributions of the low, middle, and high clouds in the CTL run"*

**Reply**: Done.

***Comment 18:***
*L371: "improves the simulations of summer precipitation in Asia" Does EXP or EXP_COR improve the precipitation in Asia? It seems like only for eastern China in EXP and TP area in EXP_COR.*

**Reply**: Thanks. We clarified this in Lines 338-340 in the revision:

"The analyses presented above demonstrate that the introduction of the subgrid heat flux schemes (EXP and EXP_COR), compared to the default model, improves the simulations of summer precipitation in eastern China in EXP and additional TP regions in EXP_COR."

***Comment 19:***
*L398-399: What values? Fig. S8 depicts the vertical structure of clouds.*

**Reply**: We apologize for the typo. It should be Fig. S10f in the revision.

***Comment 20:***
*L433-435: "Compared with MAM4 ... cloud macrophysics schemes in CAM6" Needs to be clarified. Are the authors comparing EXP_COR with physics parameterizations or comparing the physics parameterizations from different CESM versions?*

**Reply**: We are comparing EXP_COR with physics parameterizations. This has been clarified in Lines 400-403 in the revision:

"However, compared with the additional computational costs of the four-mode version

of the Modal Aerosol Module (MAM4) updated from MAM3 and the Cloud Layers Unified by Binormals (CLUBB) scheme instead of the CAM5 boundary layer turbulence, shallow convection, and cloud macrophysics schemes, respectively in CESM2 (CESM version 2), the increased computational cost in EXP_COR relative to CTL is much smaller and thus acceptable."

***Comment 21:***
*Have a native speaker assist with the writing*

**Reply**: In the original manuscript, the native English speaker has polished the language. In this round of revision, we went through the manuscript carefully again for language polishing.